# SODA: Robust Training of Test-Time Data Adaptors

**Zige Wang**[1,2]    **Yonggang Zhang**[2]    **Zhen Fang**[3]    **Long Lan**[4]
**Wenjing Yang**[4*]   **Bo Han**[2]
[1]School of Computer Science, Peking University    [2]Hong Kong Baptist University
[3]University of Technology Sydney    [4]National University of Defense Technology

## Abstract

Adapting models deployed to test distributions can mitigate the performance degradation caused by distribution shifts. However, privacy concerns may render model parameters inaccessible. One promising approach involves utilizing zeroth-order optimization (ZOO) to train a data adaptor to adapt the test data to fit the deployed models. Nevertheless, the data adaptor trained with ZOO typically brings restricted improvements due to the potential corruption of data features caused by the data adaptor. To address this issue, we revisit ZOO in the context of test-time data adaptation. We find that the issue directly stems from the unreliable estimation of the gradients used to optimize the data adaptor, which is inherently due to the unreliable nature of the pseudo-labels assigned to the test data. Based on this observation, we propose pseudo-label-robust data adaptation (SODA) to improve the performance of data adaptation. Specifically, SODA leverages high-confidence predicted labels as reliable labels to optimize the data adaptor with ZOO for label prediction. For data with low-confidence predictions, SODA encourages the adaptor to preserve data information to mitigate data corruption. Empirical results indicate that SODA can significantly enhance the performance of deployed models in the presence of distribution shifts without requiring access to model parameters.

## 1   Introduction

Deep neural networks have emerged as a dominant tool in solving artificial intelligence problems due to their exceptional performance across various tasks [19, 20, 42], thus being deployed to various environments. However, in practice, these models typically suffer performance degradation due to distribution discrepancies between training and test data [1, 13, 47, 35]. To mitigate this issue, Test-Time Adaptation (TTA) is proposed as a promising solution, where unlabeled test data are leveraged to modify the parameters of deployed models to alleviate performance degradation [7, 21, 39].

In practice, the parameters of deployed models may be unmodifiable and inaccessible in many applications due to intellectual property protection, misuse prevention, or privacy concerns in healthcare and finance [22, 46]. The difficulty in modifying model parameters has hindered previous efforts aimed at adapting deployed models to the test distribution [15, 23, 29]. It is shown that training a data adaptor to modify test data offers an alternative solution to mitigate the performance degradation caused by distribution discrepancy [18, 37]. However, the difficulty in accessing model parameters makes gradient computation a challenging task, hindering these data adaptation methods.

One straightforward approach involves utilizing zeroth-order optimization (ZOO) [25] to train the data adaptor to adapt test data to fit deployed models. In particular, ZOO can be employed to estimate gradients for the optimization of data adaptors without modifying and accessing the parameters of deployed models. Therefore, test-time data adaptation with ZOO makes it possible to improve the performance of deployed models in many practical scenarios. However, introducing data adaptors

---

[*]Corresponding Author: Wenjing Yang(wenjing.yang@nudt.edu.cn)

37th Conference on Neural Information Processing Systems (NeurIPS 2023).

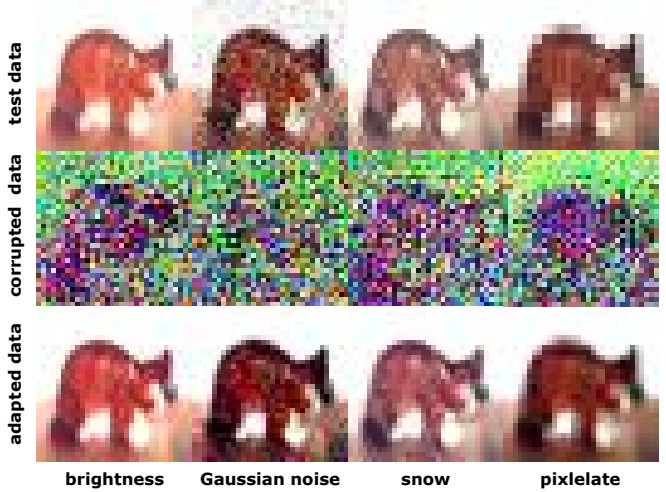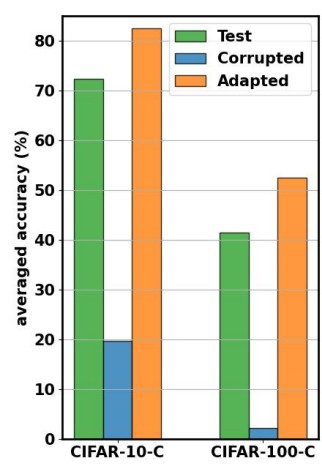

**brightness**     **Gaussian noise**     **snow**     **pixlelate**

Figure 1: Demonstration of corrupted data and adapted data. The left part shows examples of the original test data from CIFAR-10-C, the corrupted data generated by the data adaptor trained with unreliable pseudo-labels, and the adapted data generated by the data adaptor in SODA. The right part shows the corresponding accuracy of test data, corrupted data, and adapted data on CIFAR-10/100-C.

trained with ZOO brings limited improvement. To endow data adaptors with reliability in improving model performance, we revisit ZOO in the context of test-time data adaptation. Training of the data adaptor depends heavily on the predicted labels. Thus, unreliable predicted labels will lead to unreliable gradient estimations in ZOO, which makes data features corrupted rather than adapted to deployed models. This is consistent with our observations as depicted in Figure 1.

Built upon the observation, we propose pseudo-label-robust data adaptation (SODA) to enhance deployed models with inaccessible parameters. SODA robustifies the training of data adaptors by splitting the test dataset into two subsets. One subset contains data with high-confidence predictions and is regarded as a reliable dataset. The other contains the remaining data points, considered to be an unreliable subset. For data in the reliable dataset, SODA trains the data adaptor using ZOO in a supervised manner. In the meantime, SODA encourages the data adaptor to preserve input information in an unsupervised manner for data corruption mitigation over unreliable data. To verify the efficacy of SODA, we evaluate it on three widely used benchmark datasets under various settings. Our experimental results demonstrate that SODA can effectively mitigate the performance degradation of deployed models in the presence of distribution shifts.

To sum up, our main contributions are presented as follows:

- To enhance deployed models with unmodifiable and inaccessible parameters, we propose to shift from model adaptation to data adaptation, where zeroth-order optimization is employed to estimate gradients for the training of the data adaptor. Unfortunately, test-time data adaptation causes corruption of data features, leading to limited performance improvements.

- To exploit the potential of test-time data adaptation, we find that the limited improvement issue stems from the unreliable nature of the predicted labels used in ZOO. Thus, we propose SODA to robustify the training of test-time data adaptation, where the data adaptor is trained to preserve input information for data with unreliable predictions.

- We verify the effectiveness of SODA under various settings. Our experimental results demonstrate that SODA can directly enhance the deployed model under distribution shifts without accessing and modifying the model parameters.

## 2 Related Work

**Test-time adaptation.** Test-time adaptation is a machine learning technique that addresses the distribution shift problem using only unlabeled test data before making predictions. To mitigate the distribution discrepancy, most previous works adapt the pre-trained model to test data by modifying

the full or part of the model parameters. Some advanced works [7, 21] adapt the feature extractor to obtain more efficient feature representations of test data, while others [16, 17, 45] modify or replace the last linear classification layer to improve prediction over the extracted features. Batch normalization calibration [23, 34, 43, 49] is also exploited to adjust the statistics and affine parameters in batch normalization layers. Unlike previous works, our work focuses on situations where model parameters are unmodifiable and adapts test data to the pre-trained model as an alternative solution.

**White-box input adaptation.** Recently, several works have put effort into input-level optimization in test-time adaptation by changing the input data or features. Auxiliary auto-encoders [12, 18], amplitude features and Fourier-style calibration [50], label-preserving features along with a generative model [31], and generative diffusion model [10] are utilized to achieve input-level adaptation. These works require either modification of the model training process or gradients from the deployed model at test time. In contrast, our work focuses on test-time data adaptation without accessing the training process and gradients of the deployed model, making it more practical and broadly applicable.

**Domain adaptation of black-box predictors.** Domain Adaptation of Black-Box Predictors (DABP) is a subcategory of unsupervised domain adaptation [9, 30] that solves a more restricted and practical application setting. Under this setting, the pre-trained model is treated as a black box, and only the model's output is available during adaptation. Few works [22, 32, 46] have proposed solutions to this challenge using knowledge distillation to transfer knowledge from the black-box model to target models. In this work, we address the black-box setting in the context of test-time adaptation and propose to perform data adaptation without knowledge transfer. Besides, while previous DABP works require training on the entire unlabeled test dataset, our proposed method can also deal with online black-box settings where the test data arrive sequentially.

**Zeroth-order optimization.** Zeroth-order optimization (ZOO) [25] is a family of optimization methods that do not require gradient information to search for the optimal solution. Instead, it explores the searching space using various techniques [6, 24] to estimate the optimization direction. Although it can be less efficient than first-order optimization methods, ZOO methods can be helpful in scenarios where gradient information is unavailable or expensive to compute. For instance, recent studies have applied ZOO to perform adversarial attacks to black-box neural networks [48], hyperparameter optimization in federated learning without gradient descent [51], and transfer learning on black-box models [40]. In this work, we propose to leverage ZOO for test-time data adaptation on deployed models with inaccessible gradient information.

## 3 Methodology

We mainly focus on the $C$-way image classification task with a distribution shift between the training and test data, following previous works [26, 39, 43]. Given a deployed model $\mathbf{M}$ with inaccessible parameters, our purpose is to improve its prediction accuracy on unlabeled test data $\mathbf{X} = \{\mathbf{x}_1, ..., \mathbf{x}_n\}$. Since the parameters and inner structures of $\mathbf{M}$ are unknown, only the output prediction probabilities are available from $\mathbf{M}$ during the entire adaptation process. Namely, the proposed method pseudo-label-robust data adaptation (SODA) aims to adapt $\mathbf{X}$ to $\mathbf{M}$ without requiring access to the parameters of $\mathbf{M}$. The overall framework of SODA is shown in Figure 2.

### 3.1 Preliminary

Before elaborating on our proposed framework, we introduce zeroth-order optimization (ZOO). ZOO is a gradient-free alternative of first-order optimization (FOO), e.g., SGD, SCD, and Adam. Most ZOO methods follow the structure of FOO and consist of three fundamental steps [25]: gradient estimation, descent direction computation, and point updating. They utilize function-value-based gradient estimations to approximate the full or stochastic gradients computed in FOO.

One commonly used ZOO gradient estimation strategy is multi-point estimation [6, 24]. Given a continuously differentiable objective function $f(\boldsymbol{\theta})$ on a $d$-dimensional variable $\boldsymbol{\theta} \in \mathbb{R}^d$, multi-point estimation computes directional derivative approximation as follows:

$$\widehat{\nabla}_{\boldsymbol{\theta}} f(\boldsymbol{\theta}) := \frac{1}{\mu q} \sum_{i=1}^{q} \left[ (f(\boldsymbol{\theta} + \mu \mathbf{u}_i) - f(\boldsymbol{\theta})) \mathbf{u}_i \right], \tag{1}$$

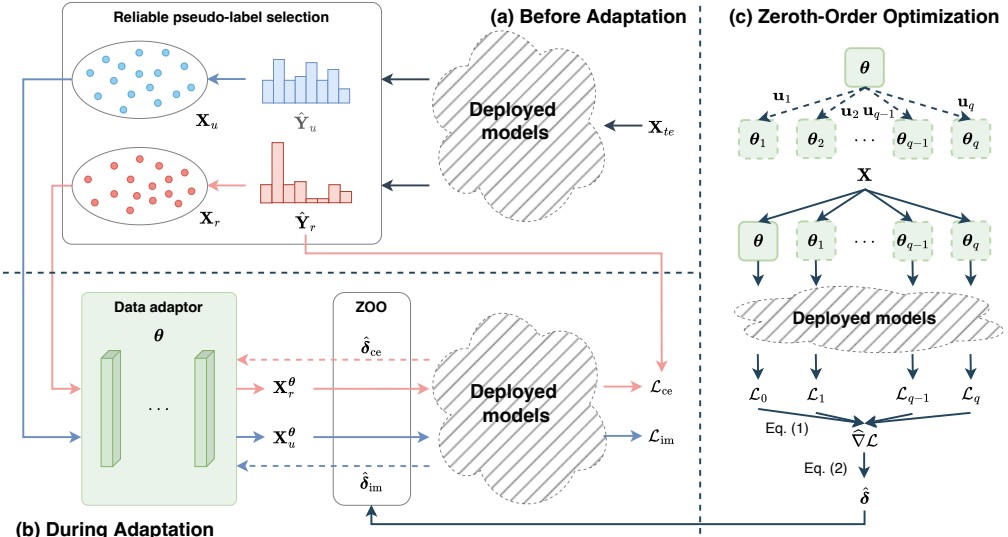

Figure 2: The overall framework of SODA. (a) Before adaptation, SODA first performs reliable pseudo-label selection according to prediction confidence. (b) During adaptation, the data adaptor with parameter $\boldsymbol{\theta}$ is trained over the test data with reliable pseudo-labels using cross-entropy loss and those with unreliable pseudo-labels using mutual information maximization. The gradient is estimated using (c) zeroth-order optimization.

where $\mathbf{u}_1, ..., \mathbf{u}_q$ are $q$ random direction vectors typically drawn from the standard multivariate normal distribution $\mathcal{N}(\mathbf{0}, \mathbf{I})$, and $\mu$ is the smoothing parameter. On a mini-batch of data points $\mathbf{x}_1, ..., \mathbf{x}_l$, the estimated gradient $\hat{\boldsymbol{\delta}}$ is the average of the multi-point estimations to all data points $\mathbf{x}_1, ..., \mathbf{x}_l$:

$$\hat{\boldsymbol{\delta}} = \frac{1}{l} \sum_{i=1}^{l} \widehat{\nabla}_{\boldsymbol{\theta}} f(\boldsymbol{\theta}; \mathbf{x}_i). \tag{2}$$

Various strategies are adopted to compute the descent direction. For zeroth-order stochastic gradient descent (ZO-SGD) utilized in our work, the descent direction is set as the current gradient estimation $\hat{\boldsymbol{\delta}}$. The point updating rule is the same as in traditional SGD: with learning rate $\eta$,

$$\boldsymbol{\theta} = \boldsymbol{\theta} - \eta \hat{\boldsymbol{\delta}}. \tag{3}$$

## 3.2 Zeroth-Order Optimization in Test-Time Data Adaptation

Let $\mathbf{G}$ with parameters $\boldsymbol{\theta}$ be the data adaptor for test-time data adaptation. For each test data point $\mathbf{x}_i$ $(i = 1, ..., n)$, $\mathbf{G}$ transforms it to form the adapted data $\mathbf{x}_i^{\boldsymbol{\theta}}$ for inference as follows:

$$\mathbf{x}_i^{\boldsymbol{\theta}} = \mathbf{G}(\mathbf{x}_i; \boldsymbol{\theta}). \tag{4}$$

Ideally, the data adaptor $\mathbf{G}$ should be trained by minimizing the KL divergence between the predicted probabilities of the adapted data and the true labels of test data. Typically, the training process requires back-propagating the gradients from the deployed model to the data adaptor. A challenge arises as gradient computation is infeasible for the deployed model $\mathbf{M}$ with inaccessible parameters. In this regard, ZOO provides an effective approach to estimating gradients, as discussed in the previous section. Considering the parameters $\boldsymbol{\theta}$ of the data adaptor $\mathbf{G}$ as the variables to be optimized, utilizing ZOO in test-time data adaptation is to replace the objective function $f(\boldsymbol{\theta})$ with the training objective function used to train the data adaptor $\mathbf{G}$. Assuming that the true one-hot label $\mathbf{y}_i$ of a test data point $\mathbf{x}_i$ is given, and replacing the function $f(\boldsymbol{\theta})$ in Eq. (1) with KL divergence loss $\mathcal{L}(\cdot, \cdot) := \mathrm{KL}(\cdot \| \cdot)$, the directional derivative approximation w.r.t. $\mathcal{L}(\cdot, \cdot)$ and $(\mathbf{x}_i, \mathbf{y}_i)$ is

$$\widehat{\nabla}_{\boldsymbol{\theta}} \mathcal{L}_i = \frac{1}{\mu q} \sum_{j=1}^{q} \left[ \left( \mathcal{L}(\mathbf{y}_i, \mathbf{M} \circ \mathbf{G}(\mathbf{x}_i; \boldsymbol{\theta} + \mu \mathbf{u}_j)) - \mathcal{L}(\mathbf{y}_i, \mathbf{M} \circ \mathbf{G}(\mathbf{x}_i; \boldsymbol{\theta})) \right) \mathbf{u}_j \right]. \tag{5}$$

In test-time adaptation, the true labels of test data are obviously unknown. A common strategy is to use the predicted pseudo-label $\hat{\mathbf{y}}_i$ as the substitute of the true label $\mathbf{y}_i$.

However, the pseudo-labels are unreliable due to the inaccurate model prediction under distribution shifts, causing the corrupted data features depicted in Figure 1. Let $\boldsymbol{\sigma}_i$ denote the disturbance of pseudo-label $\hat{\mathbf{y}}_i$, i.e., $\hat{\mathbf{y}}_i = \boldsymbol{\sigma}_i + \mathbf{y}_i$, and $\hat{\mathbf{p}}_i^{\boldsymbol{\theta}} = \mathbf{M} \circ \mathbf{G}(\mathbf{x}_i; \boldsymbol{\theta})$ denote the predicted probability of the adapted data point $\mathbf{x}_i^{\boldsymbol{\theta}} = \mathbf{G}(\mathbf{x}_i; \boldsymbol{\theta})$, the KL divergence loss at test point $\mathbf{x}_i$ becomes:

$$\mathcal{L}_i = -H(\mathbf{y}_i + \boldsymbol{\sigma}_i) + \mathcal{L}_{\mathrm{ce}}(\mathbf{y}_i, \hat{\mathbf{p}}_i^{\boldsymbol{\theta}}) - \boldsymbol{\sigma}_i \log \hat{\mathbf{p}}_i^{\boldsymbol{\theta}}, \tag{6}$$

where $\mathcal{L}_{\mathrm{ce}}(\cdot, \cdot)$ is the cross-entropy loss. Then, replacing $\mathbf{y}_i$ with $\hat{\mathbf{y}}_i$ in Eq. (5), the directional derivative approximation becomes

$$\widehat{\nabla}_{\boldsymbol{\theta}} \check{\mathcal{L}}_i = \widehat{\nabla}_{\boldsymbol{\theta}} \mathcal{L}_{\mathrm{ce}} + \frac{\boldsymbol{\sigma}_i}{\mu q} \sum_{j=1}^{q} \log \frac{\hat{\mathbf{p}}_i^{\boldsymbol{\theta}}}{\hat{\mathbf{p}}_i^{\boldsymbol{\theta}+\mu \mathbf{u}_j}} \mathbf{u}_j, \tag{7}$$

where $\widehat{\nabla}_{\boldsymbol{\theta}} \mathcal{L}_{\mathrm{ce}} = \frac{1}{\mu q} \sum_{j=1}^{q} \left[ \left( \mathcal{L}_{\mathrm{ce}}(\mathbf{y}_i, \hat{\mathbf{p}}_i^{\boldsymbol{\theta}+\mu \mathbf{u}_j}) - \mathcal{L}_{\mathrm{ce}}(\mathbf{y}_i, \hat{\mathbf{p}}_i^{\boldsymbol{\theta}}) \right) \mathbf{u}_j \right]$ is the ideal directional derivative approximation. The derivations of Eq. (6) and Eq. (7) are deferred to Appendix A. In Eq. (7), the last term is the disturbing term directly introduced by $\boldsymbol{\sigma}_i$, causing the difference between $\widehat{\nabla}_{\boldsymbol{\theta}} \mathcal{L}_{\mathrm{ce}}$ and the unreliable directional derivative approximation $\widehat{\nabla}_{\boldsymbol{\theta}} \check{\mathcal{L}}_i$. $\widehat{\nabla}_{\boldsymbol{\theta}} \check{\mathcal{L}}_i$ further leads to unreliable estimated gradients in Eq. (2), which hinders the optimization of $\boldsymbol{\theta}$ and the training of $\mathbf{G}$.

### 3.3 Pseudo-Label-Robust Training

A direct strategy to alleviate the impact of the disturbing term in Eq. (7) is to select pseudo-labels with small $\boldsymbol{\sigma}_i$. The selected pseudo-labels form reliable pseudo-label set $\hat{\mathbf{Y}}_r$ to train the data adaptor in a supervised manner. In particular, two basic criteria for reliable pseudo-label selection are adopted in SODA: 1) the prediction confidence should be higher than a threshold $\tau$, indicating the selected pseudo-labels have small disturbances; 2) the number of selected reliable pseudo-labels for each class should be less than $(1-\rho)n/C$ to maintain the balance among classes, where $\rho$ is the noise ratio, and $C$ is the number of classes. The test data points corresponding to the selected reliable pseudo-labels are considered as reliable data set $\mathbf{X}_r$, which is trained over with cross-entropy loss $\mathcal{L}_{\mathrm{ce}}$ and $\hat{\mathbf{Y}}_r$.

To mitigate the data corruption over the remaining test data points $\mathbf{X}_u$ with unreliable pseudo-labels, SODA trains over them in an unsupervised manner. Following previous works [21, 22], mutual information maximization [4, 36, 41] is a widely-used unsupervised loss that can encourage both global diversity and local certainty of model predictions by maximizing the mutual information between the input data sample and the predicted probabilities. Thus, it is adopted to preserve input information in $\mathbf{X}_u$ as shown in Eq. (8), where $\mathbf{x}_i^{\boldsymbol{\theta}} = \mathbf{G}(\mathbf{x}_i; \boldsymbol{\theta})$ and $\hat{\mathbf{p}}_i = \mathbf{M} \circ \mathbf{G}(\mathbf{x}_i; \boldsymbol{\theta})$.

$$\mathcal{L}_{\mathrm{im}}(\mathbf{X}_u^{\boldsymbol{\theta}}) = \mathbb{E}_{\mathbf{x}_i^{\boldsymbol{\theta}} \in \mathbf{X}_u^{\boldsymbol{\theta}}} \left[ \sum_{k=1}^{C} \hat{\mathbf{p}}_{ik} \log \hat{\mathbf{p}}_{ik} \right] - \sum_{k=1}^{C} \mathbb{E}_{\mathbf{x}_i^{\boldsymbol{\theta}} \in \mathbf{X}_u^{\boldsymbol{\theta}}} \hat{\mathbf{p}}_{ik} \log \mathbb{E}_{\mathbf{x}_i^{\boldsymbol{\theta}} \in \mathbf{X}_u^{\boldsymbol{\theta}}} \hat{\mathbf{p}}_{ik}. \tag{8}$$

### 3.4 Theoretical Analysis

To theoretically show the effectiveness of the proposed pseudo-label-robust training strategy, we analyze the expected gradient estimation error [5, 24] in the training of test-time data adaptor with zeroth-order optimization. For simplicity, we consider the special case where the estimated gradient equals directional derivative approximation with mini-batch size equal to 1. The expected gradient estimation error $\mathcal{R}_{\mathbf{X}}$ between the estimated gradient $\widehat{\nabla}_{\boldsymbol{\theta}} \check{\mathcal{L}}_i$ and the true gradient $\nabla_{\boldsymbol{\theta}} \mathcal{L}_i$ w.r.t. the whole test dataset $\mathbf{X}$ is:

$$\mathcal{R}_{\mathbf{X}} = \mathbb{E}_{\mathbf{X}} \left[ \mathbb{E}[\| \widehat{\nabla}_{\boldsymbol{\theta}} \check{\mathcal{L}}_i - \nabla_{\boldsymbol{\theta}} \mathcal{L}_i \|_2] \right]. \tag{9}$$

Denoting $h(\mathbf{x}_i) = -\boldsymbol{\sigma}_i \log \hat{\mathbf{p}}_i^{\boldsymbol{\theta}}$ in Eq. (6), the gradient of the KL divergence loss is $\nabla_{\boldsymbol{\theta}} \mathcal{L}_i = \nabla_{\boldsymbol{\theta}} \mathcal{L}_{\mathrm{ce}} + \nabla_{\boldsymbol{\theta}} h$. Accordingly, the estimated gradient of the KL divergence loss is $\widehat{\nabla}_{\boldsymbol{\theta}} \check{\mathcal{L}}_i = \widehat{\nabla}_{\boldsymbol{\theta}} \mathcal{L}_{\mathrm{ce}} + \widehat{\nabla}_{\boldsymbol{\theta}} h$. Then, before applying pseudo-label-robust data adaptation, the upper bound of expected gradient estimation error is:

$$\mathcal{R}_{\mathbf{X}} \le \mathbb{E}_{\mathbf{X}} \left[ \mathbb{E}[\| \widehat{\nabla}_{\boldsymbol{\theta}} \check{\mathcal{L}}_{\mathrm{ce}} - \nabla_{\boldsymbol{\theta}} \mathcal{L}_{\mathrm{ce}} \|_2] + \mathbb{E}[\| \widehat{\nabla}_{\boldsymbol{\theta}} h - \nabla_{\boldsymbol{\theta}} h \|_2] \right]. \tag{10}$$

**Algorithm 1** SODA framework

---

**Input:** test data $\mathbf{X}$, data adaptor $\mathbf{G}$ with parameters $\boldsymbol{\theta}$, deployed model $\mathbf{M}$, adaptation epochs $N$, learning rate $\eta$, $\alpha$, $q$
**STEP 1: Before adaptation:**
Compute $\hat{\mathbf{p}} = \mathbf{M}(\mathbf{X})$
Select $\hat{\mathbf{Y}}_r$, $\mathbf{X}_r$ and $\mathbf{X}_u$ as described in Section 3.3
**STEP 2: During adaptation:**
**for** 1 **to** $N$ **do**
    **for** mini-batch $\mathbf{X}_{\mathrm{b}} = \{\mathbf{x}_{\mathrm{b}_1}, \mathbf{x}_{\mathrm{b}_2}, ..., \mathbf{x}_{\mathrm{b}_l}\}$ in $\mathbf{X}$ **do**
        Compute $\hat{\mathbf{Y}}_{\mathrm{b}} = \{\hat{\mathbf{y}}_{\mathrm{b}_1}, \hat{\mathbf{y}}_{\mathrm{b}_2}, ..., \hat{\mathbf{y}}_{\mathrm{b}_l}\}$, where $\hat{\mathbf{y}}_{\mathrm{b}_i} = \mathbf{M}(\mathbf{x}_{\mathrm{b}_i})$
        Select $\hat{\mathbf{Y}}_{r_{\mathrm{b}}} = \hat{\mathbf{Y}}_r \cap \hat{\mathbf{Y}}_{\mathrm{b}} = \{\hat{\mathbf{y}}_{\mathrm{b}_1}, \hat{\mathbf{y}}_{\mathrm{b}_2}, ..., \hat{\mathbf{y}}_{\mathrm{b}_{l_r}}\}$ and their corresponding data $\mathbf{X}_{r_{\mathrm{b}}} = \{\mathbf{x}_{r_1}, \mathbf{x}_{r_2}, ..., \mathbf{x}_{r_{l_r}}\}$, the remaining data as $\mathbf{X}_{u_{\mathrm{b}}} = \{\mathbf{x}_{u_1}, \mathbf{x}_{u_2}, ..., \mathbf{x}_{u_{l_u}}\}$ $\{\mathbf{X}_{r_{\mathrm{b}}} \cup \mathbf{X}_{u_{\mathrm{b}}} = \mathbf{X}_{\mathrm{b}}\}$
        Compute $\mathbf{x}_{r_i}^{\boldsymbol{\theta}}$ and $\mathbf{x}_{u_i}^{\boldsymbol{\theta}}$ using Eq. (5) for each data point in $\mathbf{X}_{r_{\mathrm{b}}}$ and $\mathbf{X}_{u_{\mathrm{b}}}$
        Compute $\mathcal{L}_{\mathrm{ce}}$ for $\mathbf{X}_{r_{\mathrm{b}}}^{\boldsymbol{\theta}} = \{\mathbf{x}_{r_1}^{\boldsymbol{\theta}}, \mathbf{x}_{r_2}^{\boldsymbol{\theta}}, ..., \mathbf{x}_{r_{l_r}}^{\boldsymbol{\theta}}\}$ using $\hat{\mathbf{Y}}_{r_{\mathrm{b}}}$
        Compute $\mathcal{L}_{\mathrm{im}}$ for $\mathbf{X}_{u_{\mathrm{b}}}^{\boldsymbol{\theta}} = \{\mathbf{x}_{u_1}^{\boldsymbol{\theta}}, \mathbf{x}_{u_2}^{\boldsymbol{\theta}}, ..., \mathbf{x}_{u_{l_u}}^{\boldsymbol{\theta}}\}$ using Eq. (8)
        Compute estimated gradients $\hat{\boldsymbol{\delta}}_{\mathrm{ce}}$ and $\hat{\boldsymbol{\delta}}_{\mathrm{im}}$ w.r.t. $\mathcal{L}_{\mathrm{ce}}$ and $\mathcal{L}_{\mathrm{im}}$ using Eq. (5) and Eq. (2)
        $\boldsymbol{\theta} \leftarrow \boldsymbol{\theta} - \eta(\hat{\boldsymbol{\delta}}_{\mathrm{im}} + \alpha\hat{\boldsymbol{\delta}}_{\mathrm{ce}})$
    **end for**
**end for**

---

In pseudo-label-robust training strategy, the test dataset $\mathbf{X}$ is separated into reliable set $\mathbf{X}_r$ learned by cross-entropy loss with pseudo-labels, and unreliable set $\mathbf{X}_u$ learned by mutual information loss, the expected gradient estimation error w.r.t. $\mathbf{X}$ becomes:

$$\widetilde{\mathcal{R}}_{\mathbf{X}} = \mathbb{E}_{\mathbf{X}_r}\left[\mathbb{E}[\|\ \hat{\nabla}_{\boldsymbol{\theta}}\mathcal{L}_{\mathrm{ce}} - \nabla_{\boldsymbol{\theta}}\mathcal{L}_{\mathrm{ce}} + \hat{\nabla}_{\boldsymbol{\theta}}h - \nabla_{\boldsymbol{\theta}}h\ \|_2]\right] + \mathbb{E}_{\mathbf{X}_u}\left[\mathbb{E}[\|\ \hat{\nabla}_{\boldsymbol{\theta}}\mathcal{L}_{\mathrm{im}} - \nabla_{\boldsymbol{\theta}}\mathcal{L}_{\mathrm{im}}\ \|_2]\right]. \quad (11)$$

According to the previous study [3], minimizing the cross entropy loss $\mathcal{L}_{\mathrm{ce}}(\mathbf{y}_i, \hat{\mathbf{p}}_i^{\boldsymbol{\theta}})$ is equivalent to maximizing the mutual information $\mathcal{L}_{\mathrm{im}}$. Then, the upper bound of the expected gradient estimation error w.r.t. $\mathbf{X}$ is:

$$\widetilde{\mathcal{R}}_{\mathbf{X}} \le \mathbb{E}_{\mathbf{X}_r}\left[\mathbb{E}[\|\ \hat{\nabla}_{\boldsymbol{\theta}}\mathcal{L}_{\mathrm{ce}} - \nabla_{\boldsymbol{\theta}}\mathcal{L}_{\mathrm{ce}}\ \|_2] + \mathbb{E}[\|\ \hat{\nabla}_{\boldsymbol{\theta}}h - \nabla_{\boldsymbol{\theta}}h\ \|_2]\right] + \mathbb{E}_{\mathbf{X}_u}\left[\mathbb{E}[\|\ \hat{\nabla}_{\boldsymbol{\theta}}\mathcal{L}_{\mathrm{ce}} - \nabla_{\boldsymbol{\theta}}\mathcal{L}_{\mathrm{ce}}\ \|_2]\right]. \quad (12)$$

Comparing Eq. (10) and Eq. (12), the upper bound of the expected gradient estimation error is tightened after applying the pseudo-label-robust training strategy, justifying that the proposed pseudo-label-robust training strategy can alleviate the disturbance caused by the unreliable pseudo-labels in the gradient estimation in zeroth-order optimization of the test-time data adaptor.

### 3.5 SODA Framework

Up to this point, we have discussed zeroth-order optimization in test-time data adaptation and our proposed pseudo-label-robust training strategy. The overall training objective is shown in Eq. (13),

$$\mathcal{L}_{\mathrm{all}}(\mathbf{X}, \hat{\mathbf{Y}}_r) = -\mathcal{L}_{\mathrm{im}}(\mathbf{X}_u) + \alpha\mathcal{L}_{\mathrm{ce}}(\mathbf{X}_r, \hat{\mathbf{Y}}_r), \quad (13)$$

where $\alpha$ is the balancing hyperparameter between mutual information maximization loss $\mathcal{L}_{\mathrm{im}}$ and cross entropy loss $\mathcal{L}_{\mathrm{ce}}$. The training algorithm of SODA is summarized in Algorithm 1.

## 4 Experiments

### 4.1 Experimental Setup

**Datasets.** We first evaluate our proposed framework SODA on two widely used out-of-distribution benchmarks, namely **CIFAR-10-C** and **CIFAR-100-C** [14], each containing 10,000 CIFAR-10/100 test images corrupted by 19 kinds of corruptions with 5 severity levels. We further evaluate SODA on a large scale dataset **ImageNet-C** [14] with 50,000 ImageNet validation images corrupted by the same corruptions as CIFAR10/100-C. We conduct our experiments on the highest level of each corruption and report the averaged accuracies over 19 corruptions.

Table 1: Average accuracies (%) on CIFAR-10-C (**C10-C**), CIFAR-100-C(**C100-C**) and ImageNet-C (**IN-C**). **FO Grad.** means the requirement of first-order gradient from the target model or deployed model. **Model Mod.** means the requirement of modifying the parameters of deployed models. **Distill.** indicates methods using knowledge distillation to learn target models. **DA** indicates methods using test-time data adaptation. **MA** indicates methods using model adaptation.

| Categories | Methods | FO Grad. | Model Mod. | C10-C | C100-C | IN-C |
|---|---|---|---|---|---|---|
| - | Deployed | - | - | 72.39 | 41.41 | 31.36 |
| Distill. | DINE | ✓ | ✗ | 73.86 | 40.52 | - |
| | BETA | ✓ | ✗ | 75.71 | 39.62 | - |
| DA | DA-PGD | ✗ | ✗ | 24.63 | 4.15 | 14.39 |
| | DA-ZOO-Input | ✗ | ✗ | 68.70 | 31.53 | 17.57 |
| | DA-Direct | ✗ | ✗ | 70.48 | 37.67 | 29.37 |
| | DA-PL | ✗ | ✗ | 72.93 | 41.44 | 31.91 |
| | SODA (Ours) | ✗ | ✗ | **82.55** | **52.41** | **42.14** |
| | SODA-R (Ours) | ✓ | ✗ | **88.39** | 60.31 | 48.70 |
| MA | MA-SO | ✓ | ✓ | 86.54 | **62.02** | **56.90** |

**Baselines. Deployed** is the deployed model without adaptation. We compare our proposed SODA framework with two DABP baselines utilizing knowledge distillation. **DINE** [22] distills the knowledge from the deployed model to a target model by minimizing the KL divergence between smoothed pseudo-labels and target model predictions. **BETA** [46] divides the target domain into easy- and hard-to-adapt subdomains, then mutually distills twin networks with weak-strong augmentation on two subdomains. We further implement four vanilla baselines of test-time data adaptation using ZOO. **DA-Direct** directly generates adapted data instead of perturbations using the same network structure and initial pseudo-labels. **DA-PGD** adopts PGD [27] to directly generate perturbations using estimated gradients computed by ZOO and the training objective of SODA. **DA-ZOO-Input** uses the same data adaptor and training objective as SODA, except that the variables optimized in ZOO are the input data of the deployed model, i.e., the output adapted data of the data adaptor. The gradients of data adaptor parameters are computed based on the estimated gradients w.r.t. the adapted data. **DA-PL** trains the same data adaptor as SODA using initial pseudo-labels. Assuming gradient information is accessible, we also implement **SODA-R** using first-order gradients computed from the deployed model. To compare SODA with model adaptation, we implement **MA-SO** that modifies model parameters (except the last linear layer) using the same training objective as SODA.

**Deployed model settings.** For all experiments regarding CIFAR-10/100-C tasks, we adopt the CIFAR-10/100 pretrained ResNet-50 [11] model used in [26] and [38] as the deployed model. For experiments regarding the ImageNet-C task, we adopt the ImageNet pre-trained ResNet-50 model provided by TorchVision [28] as the deployed model. Except for SODA-R and MA-SO, the deployed model is frozen with only output probabilities accessible. For SODA-R, the deployed model is also frozen, but gradients can be computed and back-propagated through the model. For MA-SO, the deployed model is set in training mode, and all parameters can be modified.

**Implementation details.** For SODA, the data adaptor uses a small network with two convolutional layers and an instance normalization layer in between to generate perturbations added to the original test data. For SODA-R in the CIFAR-10/100-C tasks, two ResNet blocks as in [33] are inserted between the convolutional layers to form a larger data adaptor. For SODA-R in the ImageNet-C task, one ResNet block and a couple of downsampling/upsampling layers are inserted instead. Detailed data adaptor structure is described in Appendix B.3. For all methods except DINE, BETA, DA-PGD, and SODA-R, the data adaptor/model is optimized using SGD with learning rate = 1e-3, momentum = 0.9, and weight decay = 1e-5. For DINE and BETA, we follow the same settings as in [22] and [46], and the target models are both ResNet-50 initialized with ImageNet-pretrained weights downloaded from TorchVision [28]. For DA-PGD, the step size is also set to be 1e-3. SODA-R is optimized using Adam with learning rate = 1e-3 and weight decay = 1e-5. Batch size = 256 is fixed for all methods. The number of training epochs = 150 for all baselines except DINE and BETA. For DINE and BETA, we train them for 120 epochs and fine-tune them for 30 epochs. For all methods using ZOO, the query number $q = 5$ for CIFAR-10-C and ImageNet-C and $q = 10$ for CIFAR-100-C, smoothing parameter

Table 2: Comparison of SODA using ZOO with different query numbers and SODA using first-order gradients. The network structure, training objective, and training strategy are the same. Averaged accuracies (%) over 19 corruptions are reported.

| Query Numbers | 2 | 5 | 10 | 20 | 50 | SODA-FO |
|---|---|---|---|---|---|---|
| CIFAR-10-C | 82.43 | 82.55 | 82.57 | 82.59 | 82.53 | 85.97 |
| CIFAR-100-C | 51.03 | 52.19 | 52.41 | 52.97 | 52.97 | 54.32 |

$\mu$ = 1e-3. All experiments are repeated with three random seeds. The code implementation can be found at `https://github.com/tmlr-group/SODA`.

## 4.2 Experimental Results

**Effectiveness of SODA.** Table 1 shows the accuracies on CIFAR-10-C, CIFAR-100-C and ImageNet-C averaged over 19 corruptions. In the mostly restricted setting where first-order gradient computation and model modification are both not allowed, SODA improves the deployed model prediction accuracy by a large margin, 10% on CIFAR-10-C, 11% on CIFAR-100-C and 11% on ImageNet-C. SODA also outperforms the DABP baselines on CIFAR10-C and CIFAR100-C tasks. Note that DINE and BETA are excluded in experiments on ImageNet-C because they are required to initialize their target models by ImageNet pre-trained weights, which violates the TTA setting. Especially on CIFAR-100-C and ImageNet-C task with much lower initial prediction accuracy, all baselines except MA-SO fail to improve, while SODA and SODA-R make significant improvement.

**SODA v.s. model adaptation.** Further relaxing the restriction on the deployed model, when parameters of the deployed model are frozen, but gradient computation is feasible, SODA-R achieves comparable accuracy with completely unrestricted MA-SO on CIFAR-100-C and even better accuracy on CIFAR-10-C. It shows that data adaptation can be as effective as model adaptation for deployed models with inaccessible parameters. On ImageNet-C, SODA-R performs worse than MA-SO, suggesting that improvement is still needed on large-scale datasets.

**Effect of pseudo-label-robust training strategy in SODA.** The failures of DA-Direct show that directly generating data using unreliable pseudo-labels fails to adapt data to the deployed model due to the corrupted data features. By adding perturbations to the original test data, DA-PL improves from DA-Direct to a small degree but still fails to enhance the deployed model. With the same data adaptor as DA-PL, SODA can improve the model prediction to a large degree, indicating the effectiveness of the proposed pseudo-label-robust training strategy.

**Effect of data adaptor parameter ZOO in SODA.** Both using a perturbation generation strategy, DA-PGD fails by directly generating perturbations using estimated gradients without learning a data adaptor, and DA-ZOO-Input achieves worse results by directly optimizing the model input using ZOO. With the same training objective, the significant improvements made by SODA indicate the effectiveness of the data adaptor parameter ZOO adopted in SODA.

## 4.3 Discussion

**Effect of query number in ZOO.** To analyze the relation between query number $q$ and adaptation accuracy in SODA, we conduct experiments with $q \in \{2, 5, 10, 20, 50\}$. The results in Table 2 show that SODA is not sensitive to query number used in ZOO. Especially on CIFAR-10-C, only a slight performance drop is observed when $q = 2$. On CIFAR-100-C, a larger query number contributes to accuracy improvement more observably, but when $q > 10$, the contribution becomes less efficient, as the computation cost is also increased. Balancing the trade-off between accuracy and computation costs, we finally choose $q = 5$ for CIFAR-10-C and 10 for CIFAR-100-C.

**Zeroth-order optimization v.s. first-order optimization.** To better illustrate the effect of ZOO, we implement a comparing baseline as SODA-FO that shares the same setting with SODA but uses the first-order gradients back-propagated from the deployed model. Note that SODA-FO and SODA-R differ in data adaptor network structure, optimizer, and other strategies as discussed in Appendix C.1. As shown in Table 2, although SODA does not achieve the same accuracies as SODA-FO due to the unavoidable gradient estimation error, it still indicates competitive performance when gradient computation is infeasible. In Figure 3, SODA using ZOO has the slowest convergence speeds

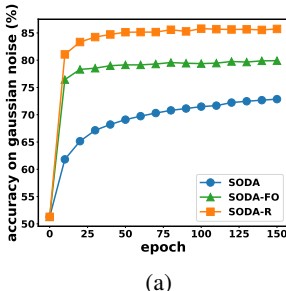

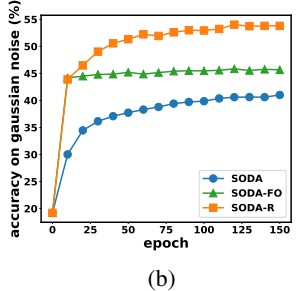

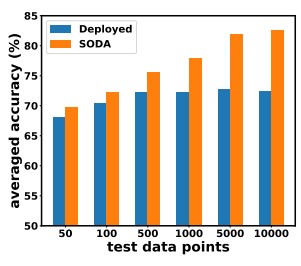

|     (a)     |     (b)     |

Figure 3: Accuracy convergence on Gaussian noise in (a) CIFAR-10-C and (b) CIFAR-100-C.

Figure 4: Comparison of SODA trained over different numbers of test samples.

Table 3: Comparison of SODA and SODA-R using data adaptor with different numbers of ResNet blocks. Averaged accuracies (%) over 19 corruptions on CIFAR-10-C are reported.

| #blocks | 0 | 1 | 2 | 3 |
|---------|-------|-------|-------|-------|
| SODA    | 82.55 | 81.91 | 80.56 | 78.65 |
| SODA-R  | 86.07 | 88.07 | 88.39 | 88.51 |

Table 4: Average accuracies (%) on CIFAR-10-C (**C10-C**) and CIFAR-100-C (**C100-C**) under online setting with difference batch size (epochs = 10). **D** is the deployed model, **BS** is batch size.

| Methods | D | SODA-O | | | |
|---------|-------|-------|-------|-------|-------|
| BS      | -     | 32    | 64    | 128   | 256   |
| C10-C   | 72.39 | 78.01 | 78.66 | 78.79 | 77.03 |
| C100-C  | 41.41 | 45.82 | 47.11 | 47.26 | 45.93 |

and lowest accuracies on both datasets. SODA-FO has similar convergence speeds as SODA-R but converges to lower accuracies, indicating that the strategies used in SODA-R with first-order optimization, i.e., deeper network and Adam optimizer, can boost the training of the data adaptor.

**Effect of data adaptor network complexity.** We also explore the effect of data adaptor network complexity on SODA and SODA-R. To increase network complexity, we change the number of ResNet blocks in the data adaptor. Results of data adaptor with {0, 1, 2, 3} ResNet blocks are reported in Table 3. For SODA, accuracy decreases as network complexity increases, indicating that complex networks hinder data adaptation using ZOO. However, the accuracy of SODA-R increases along with network complexity. This contrast illustrates that a more complex data adaptor can achieve higher accuracy but is restricted and even encumbered by zeroth-order gradient estimation.

**SODA with fewer test data points.** We further explore the effectiveness of SODA with fewer test data points. We randomly choose {50, 100, 500, 1000, 5000, 10000} test data points evenly distributed across 10 classes in CIFAR-10-C for each corruption as smaller test datasets. The averaged accuracies are reported in Figure 4. The performance of SODA is better when training data adaptor over more test data points. Nevertheless, training over 5,000 data points achieves comparable accuracy with training over 10,000 data points, showing that SODA does not require an extremely large number of data points to achieve good performance. Besides, SODA can still improve with less than 500 test data points, providing promising insights for test-time data adaption with smaller test datasets.

**Hyperparameter sensitivity.** We conduct sensitivity analysis regarding the noise ratio $\rho$, the confidence threshold $\tau$ and the balancing parameter $\alpha$. The detailed results illustrated in Appendix C.3 show that SODA is robust to different combinations of hyperparameters, but adapting an adapted threshold instead of a fixed threshold might further improve the performance of SODA.

## 4.4   SODA for Online Test-Time Adaptation

More practically, test data is not entirely available but arrives sequentially, i.e., online test-time adaptation. Hence, we further implement SODA-O as a variant of SODA under online settings. Given mini-batches of test data arrived sequentially $\{\mathbf{X}_1, ..., \mathbf{X}_T\}$, SODA-O adapts one mini-batch at a time before processing the next mini-batch with knowledge accumulated from the previous mini-batches.

The main difference between SODA-O and SODA lies in the reliable pseudo-label selection. Without access to the test data before adaptation, SODA-O maintains an ordered queue $\mathbf{Q}$ with maximum size $S$ to store the selected reliable pseudo-labels $\mathbf{Y}_r$ and their corresponding data points $\mathbf{X}_r$. Specifically,

Table 5: Average accuracies (%) on CIFAR-10-C and CIFAR-100-C under online setting with different number of epochs per batch (batch size = 256).

| Methods | Deployed | SODA-O | | | | | | SODA |
|---|---|---|---|---|---|---|---|---|
| Epochs/Batch | - | 5 | 10 | 30 | 50 | 100 | 150 | 150* |
| CIFAR-10-C | 72.39 | 75.22 | 77.03 | 79.63 | 80.38 | 81.33 | 81.71 | 82.55 |
| CIFAR-100-C | 41.41 | 43.59 | 45.81 | 48.56 | 49.26 | 50.04 | 50.12 | 52.41 |

*SODA is trained over the entire test dataset for 150 epochs

for a mini-batch $\mathbf{X}_t$ arrives at time $t$, reliable pseudo-labels $\mathbf{Y}_{r_t}$ with prediction confidences higher than $\tau$ are selected and pushed into $\mathbf{Q}$ along with their corresponding test data points $\mathbf{X}_{r_t}$. To maintain the class balance in $\mathbf{Q}$, the pseudo-labels with the smallest confidence for class $k$ will be popped out once the number of pseudo-labels for $k$ in $\mathbf{Q}$ is larger than $S/C$. Then, the remaining data points $\mathbf{X}_{u_t}$ in $\mathbf{X}_t$ are considered as $\mathbf{X}'_{u_t}$, all pseudo-labels and data points stored in $\mathbf{Q}$ are considered as $\mathbf{Y}'_{r_t}$ and $\mathbf{X}'_{r_t}$. $\mathbf{X}'_{u_t}$ and $\mathbf{X}'_{r_t}$ form a small test dataset to train the data adaptor as in SODA, i.e. Step 2 in Algorithm 1. After adaptation, the inference of $\mathbf{X}_t$ is given by the current data adaptor and the deployed model. Note that the optimization in SODA-O is not repeated after reaching the entire test dataset but only repeats for the current test data batch and the cached queue. During the adaptation of the current test data batch, the previous data batches are no longer available except for those saved in the queue. The data adaptor, hyperparameter setting, and training strategy of SODA-O are the same as SODA. The queue size $S = 1,000$ for both CIFAR-10-C and CIFAR-100-C.

**Effect of epochs per mini-batch.** We fix batch size = 256 and conduct experiments with different numbers of epochs per mini-batch, i.e., {5, 10, 30, 50, 100, 150} epochs/batch. As shown in Table 5, SODA-O is effective under online setting. As epochs/batch increase, the accuracy of SODA-O also increases and approaches the accuracy of SODA. But more training epochs means more processing time for each mini-batch, leading to a time-accuracy trade-off. With 10 epochs/batch, SODA-O can still improve the deployed model by 5% and 4% on CIFAR-10-C and CIFAR-100-C.

**Effect of batch size.** Fixing epochs/batch = 10, we also conduct experiments with different batch sizes, {32, 64, 128, 256}. The averaged accuracies are shown in Table 4. The results show that the performance of SODA-O is stable across different batch sizes. The best performance is achieved with batch size = 128. Only a slight performance drop is observed when batch size is reduced to 32, indicating that SODA-O can handle relatively small batch size.

## 5   Limitations

While providing valuable insights into robust test-time data adaptation for deployed models with inaccessible parameters, it is also essential to consider the limitations that may have affected the efficiency and effectiveness of our proposed framework. The main limitation of SODA is the time consumption brought by i) multiple queries required for gradient estimation in ZOO and ii) multiple passes required for training the random initialized data adaptor. In the future, one solution to alleviate this issue could be leveraging ZOO methods with higher query efficiency and less estimation error. Moreover, it is unclear whether SODA designed for OOD generalization can benefit OOD detection [8, 44]. Thus, we leave the exploration for OOD detection as our future work.

## 6   Conclusion

This paper focuses on two major challenges in adapting deployed machine learning models with inaccessible parameters at test-time: unmodifiable model parameters and infeasible gradient computation. Without modifying the model parameters, a data adaptor is adopted to adapt test data to the deployed model. Zeroth-order optimization is further leveraged to train the data adaptor with estimated gradients. Revisiting ZOO in test-time data adaptation, we discover that the unreliable gradient estimation in ZOO is due to the unreliable pseudo-labels assigned to test data. The proposed pseudo-label-robust data adaptation (SODA) addresses this issue with reliable pseudo-label selection and input information maximization. Our experiments on three widely used out-of-distribution benchmarks demonstrate the effectiveness of SODA in both offline and online settings.

## Acknowledgments and Disclosure of Funding

This work was partially supported by the National Natural Science Foundation of China (No. 62372459, No. 62376282).

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

# Appendix

## A  Derivation of Directional Derivative Approximation in SODA

In Section 3.2, given a deployed model $\mathbf{M}$, the ideal objective function of training the data adaptor $\mathbf{G}$ with parameters $\boldsymbol{\theta}$ in SODA is the KL divergence between the predicted probability $\hat{\mathbf{p}}_i^{\boldsymbol{\theta}} = \mathbf{M} \circ \mathbf{G}(\mathbf{x}_i; \boldsymbol{\theta})$ of the adapted data point $\mathbf{x}_i^{\boldsymbol{\theta}} = \mathbf{G}(\mathbf{x}_i; \boldsymbol{\theta})$ and the true label $\mathbf{y}_i$ of the original data point $\mathbf{x}_i$. Because $\mathbf{y}_i$ is not available at test time, pseudo-label $\hat{\mathbf{y}}_i$ predicted by $\mathbf{M}$ is adopted as a substitute of $\mathbf{y}_i$. Due to the inaccurate model prediction under distribution shifts, there is a disturbance $\boldsymbol{\sigma}_i$ in $\hat{\mathbf{y}}_i$ compared to $\mathbf{y}_i$, i.e. $\hat{\mathbf{y}}_i = \boldsymbol{\sigma}_i + \mathbf{y}_i$. Hence, the KL divergence loss $\mathcal{L}(\cdot, \cdot) := KL(\cdot \| \cdot)$ at test data point $\mathbf{x}_i$ is

$$
\begin{aligned}
\mathcal{L}_i &= KL(\hat{\mathbf{y}}_i \| \hat{\mathbf{p}}_i^{\boldsymbol{\theta}}) = \hat{\mathbf{y}}_i \log \frac{\hat{\mathbf{y}}_i}{\hat{\mathbf{p}}_i^{\boldsymbol{\theta}}} \\
&= (\mathbf{y}_i + \boldsymbol{\sigma}_i) \log \frac{\mathbf{y}_i + \boldsymbol{\sigma}_i}{\hat{\mathbf{p}}_i^{\boldsymbol{\theta}}} \\
&= (\mathbf{y}_i + \boldsymbol{\sigma}_i) \log(\mathbf{y}_i + \boldsymbol{\sigma}_i) - \mathbf{y}_i \log \hat{\mathbf{p}}_i^{\boldsymbol{\theta}} - \boldsymbol{\sigma}_i \log \hat{\mathbf{p}}_i^{\boldsymbol{\theta}} \\
&= -H(\mathbf{y}_i + \boldsymbol{\sigma}_i) + \mathcal{L}_{\mathrm{ce}}(\mathbf{y}_i, \hat{\mathbf{p}}_i^{\boldsymbol{\theta}}) - \boldsymbol{\sigma}_i \log \hat{\mathbf{p}}_i^{\boldsymbol{\theta}}
\end{aligned}
\tag{14}
$$

where $\mathcal{L}_{\mathrm{ce}}(\mathbf{y}_i, \hat{\mathbf{p}}_i^{\boldsymbol{\theta}})$ is the cross entropy loss between $\mathbf{y}_i$ and $\hat{\mathbf{p}}_i^{\boldsymbol{\theta}}$. Because the gradient information is inaccessible from the deployed model, zeroth-order optimization (ZOO) is utilized to estimate gradients for the training of the data adaptor in SODA. To do this, the objective function $f(\boldsymbol{\theta})$ in Eq. (1) is replaced with the training objective function $\mathcal{L}_i$ in test-time data adaptation. Denote $\mathcal{L}_i^{\boldsymbol{\theta}}$ as the KL divergence loss computed by data adaptor with parameters $\boldsymbol{\theta}$, the directional derivative approximation of ZOO is

$$
\begin{aligned}
\widehat{\nabla}_{\boldsymbol{\theta}} \check{\mathcal{L}}_i &= \frac{1}{\mu q} \sum_{j=1}^{q} \left[ \left( KL(\hat{\mathbf{y}}_i \| \hat{\mathbf{p}}_i^{\boldsymbol{\theta}+\mu\mathbf{u}_j}) - KL(\hat{\mathbf{y}}_i \| \hat{\mathbf{p}}_i^{\boldsymbol{\theta}}) \right) \mathbf{u}_j \right] \\
&= \frac{1}{\mu q} \sum_{j=1}^{q} \left[ \left( (\mathcal{L}_{\mathrm{ce}}(\mathbf{y}, \hat{\mathbf{p}}_i^{\boldsymbol{\theta}+\mu\mathbf{u}_j}) - \boldsymbol{\sigma}_i \log \hat{\mathbf{p}}_i^{\boldsymbol{\theta}+\mu\mathbf{u}_j}) - (\mathcal{L}_{\mathrm{ce}}(\mathbf{y}, \hat{\mathbf{p}}_i^{\boldsymbol{\theta}}) - \boldsymbol{\sigma}_i \log \hat{\mathbf{p}}_i^{\boldsymbol{\theta}}) \right) \mathbf{u}_j \right] \\
&= \frac{1}{\mu q} \sum_{j=1}^{q} \left[ (\mathcal{L}_{\mathrm{ce}}(\mathbf{y}, \hat{\mathbf{p}}_i^{\boldsymbol{\theta}+\mu\mathbf{u}_j}) - \mathcal{L}_{\mathrm{ce}}(\mathbf{y}, \hat{\mathbf{p}}_i^{\boldsymbol{\theta}})) \mathbf{u}_j \right] + \frac{1}{\mu q} \sum_{j=1}^{q} \left[ (\boldsymbol{\sigma}_i \log \hat{\mathbf{p}}_i^{\boldsymbol{\theta}} - \boldsymbol{\sigma}_i \log \hat{\mathbf{p}}_i^{\boldsymbol{\theta}+\mu\mathbf{u}_j}) \mathbf{u}_j \right] \\
&= \widehat{\nabla}_{\boldsymbol{\theta}} \mathcal{L}_{\mathrm{ce}} + \frac{\boldsymbol{\sigma}_i}{\mu q} \sum_{j=1}^{q} \log \frac{\hat{\mathbf{p}}_i^{\boldsymbol{\theta}}}{\hat{\mathbf{p}}_i^{\boldsymbol{\theta}+\mu\mathbf{u}_j}} \mathbf{u}_j,
\end{aligned}
\tag{15}
$$

where $\widehat{\nabla}_{\boldsymbol{\theta}} \mathcal{L}_{\mathrm{ce}} = \frac{1}{\mu q} \sum_{j=1}^{q} \left[ (\mathcal{L}_{\mathrm{ce}}(\mathbf{y}_i, \hat{\mathbf{p}}_i^{\boldsymbol{\theta}+\mu\mathbf{u}_j}) - \mathcal{L}_{\mathrm{ce}}(\mathbf{y}_i, \hat{\mathbf{p}}_i^{\boldsymbol{\theta}})) \mathbf{u}_j \right]$ is the ideal directional derivative approximation.

## B  Implementation Details

### B.1  Implementation details of DINE and BETA

The implementations of DINE and BETA on CIFAR-10-C and CIFAR-100-C are kept the same, following their original work [22] and [46]. For DINE, the momentum hyperparameter $\gamma = 0.6$ and the Mixup balancing hyperparameter $\beta = 1$. For BETA, $\tau = 0.8$ for domain division, $\alpha = 1.0$ for Mixup, $\lambda_{mse} = 0$, sharpening factor $T = 0.5$, and adversarial regularizer $\gamma = 0.1$. The training strategy of DINE and BETA are both SGD with learning rate = 0.001 for target network backbones and 0.01 for MLP classifiers. momentum = 0.9 and weight decay = 1e-3 are also adopted.

### B.2  Software and hardware

In our paper, all models are implemented using PyTorch 1.13.1. The ImageNet pre-trained weights used in DINE and BETA are downloaded from TorchVision 0.14.1. The experiments are conducted using NVIDIA A100-PCIE-40GB GPU with CUDA 11.7.

### B.3 Network structure of data adaptor

Figure 5 shows the network structure of the data adaptor used in our experiments. The basic structure of the data adaptor consists of two convolutional layers and an instance normalization layer in between. Multiple ResNet blocks and downsampling/upsampling layers can be inserted into the convolutional layers to form a deeper network as in [33]. For all methods except DA-Direct, the adapted data is generated by treating the network output as perturbation and adding it to the original data.

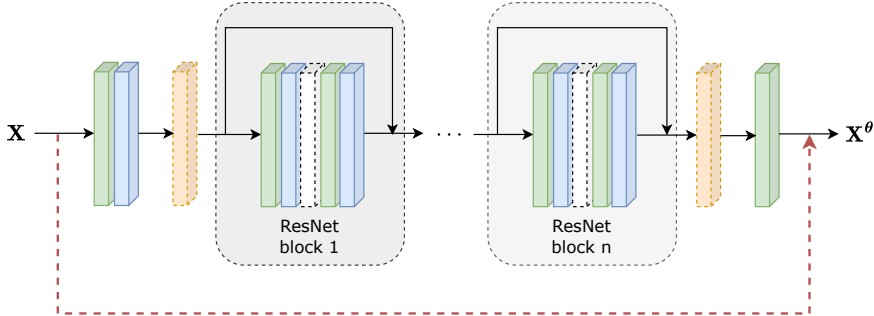

Figure 5: Network structure of data adaptor. The green block is the convolutional layer, the blue block is the instance normalization layer, the white block is the dropout layer, and the orange block is the downsampling/upsampling layer. The dashed blocks can be added or removed to form different network structures. The red dashed line means the network output is added to the original data to generate the adapted data.

## C  Additional Analysis

### C.1  Discussion about SODA and SODA-R

Compared with SODA, SODA-R uses computed first-order gradients and adopts several techniques to improve the performance, i.e., deeper data adaptor with 2 ResNet blocks, Adam optimizer, perturbation regularization, and dropout. The effect of network complexity has already been discussed in Section 4.3. In this subsection, we first introduce the perturbation regularization used in SODA-R, then evaluate the effect of perturbation regularization, different optimizers, and dropout on SODA and SODA-R.

#### C.1.1  Perturbation regularization in SODA-R

In SODA and SODA-R, the adapted data is computed by perturbing the original data with a generated perturbation. To further restrict the impact of generated perturbations on data $\mathbf{X}_r = \{\mathbf{x}_{r_1}, \mathbf{x}_{r_2}, ..., \mathbf{x}_{r_{l_r}}\}$ and $\mathbf{X}_u = \{\mathbf{x}_{u_1}, \mathbf{x}_{u_2}, ..., \mathbf{x}_{u_{l_u}}\}$, perturbation regularization with $l_1$ norm is used: let $\mathbf{x}_{r_i}^{\boldsymbol{\theta}}$ and $\mathbf{x}_{u_i}^{\boldsymbol{\theta}}$ be the corresponding adapted data of $\mathbf{x}_{r_i}$ and $\mathbf{x}_{r_u}$,

$$\mathcal{R}(\mathbf{X}) = \mathbb{E}_{\mathbf{x}_i \in \mathbf{X}_r} \big\| \mathbf{x}_{r_i}^{\boldsymbol{\theta}} - \mathbf{x}_{r_i} \big\|_1 + \mathbb{E}_{\mathbf{x}_i \in \mathbf{X}_u} \big\| \mathbf{x}_{u_i}^{\boldsymbol{\theta}} - \mathbf{x}_{u_i} \big\|_1. \tag{16}$$

First-order gradients of the perturbation regularization are directly computed and back-propagated through the data adaptor. Hence, the training objective of SODA-R becomes:

$$\mathcal{L}_{\text{all}}(\mathbf{X}, \hat{\mathbf{Y}}_r) = \mathcal{L}_{\text{im}}(\mathbf{X}_u) + \alpha \mathcal{L}_{\text{ce}}(\mathbf{X}_r, \hat{\mathbf{Y}}_r) + \beta \mathcal{R}(\mathbf{X}), \tag{17}$$

where $\beta$ is the weight of perturbation regularization and is set to be 0.005 for CIFAR-10-C and 0.01 for CIFAR-100-C.

#### C.1.2  Evaluation of perturbation regularization in SODA and SODA-R

We evaluate the effect of perturbation regularization in SODA and SODA-R on CIFAR-10-C and CIFAR-100-C. Except for the perturbation regularization term in the training objective, all other settings are kept the same as in the main experiments. The results are shown in Table 6 and Table 7.

It shows that perturbation regularization can improve the performance of SODA-R using first-order optimization, especially on CIFAR-100-C. However, it largely hinders the performance of SODA using zeroth-order optimization. The computed first-order gradients of perturbation regularization are more accurate than the estimated zeroth-order gradients of the main training objective. Thus, the data adaptor tends to optimize the perturbation regularization term first, resulting in perturbations with too small norms. The perturbations with too small norms do not have enough ability to modify the test data, which might be the reason for the worse performance achieved by SODA with perturbation regularization. One possible solution to this problem could be treating perturbation regularization as an optimization constraint and using constrained ZOO methods to train the data adaptor.

Table 6: Comparison of SODA and SODA-R with and without perturbation regularization on CIFAR-10-C. $\beta = 0.005$ in experiments w/ regularization.

| Methods | SODA | SODA-R |
|---|---|---|
| w/ regularization | 73.40 | 88.39 |
| w/o regularization | 82.55 | 87.96 |

Table 7: Comparison of SODA and SODA-R with and without perturbation regularization on CIFAR-100-C. $\beta = 0.01$ in experiments w/ regularization.

| Methods | SODA | SODA-R |
|---|---|---|
| w/ regularization | 42.27 | 60.31 |
| w/o regularization | 52.41 | 58.11 |

### C.1.3 Evaluation of optimizers in SODA and SODA-R

We evaluate the effect of optimizers used in SODA and SODA-R on CIFAR-10-C and CIFAR-100-C. Except for the optimizer used to train the data adaptor, all other settings are kept the same as in the main experiments. The results are shown in Table 8 and Table 9. On CIFAR-10-C, SODA trained by SGD and Adam achieve almost the same accuracy, while SODA-R trained by Adam achieves 3.3% higher accuracy than SGD. On CIFAR-100-C, SODA-R trained by Adam still outperforms SODA-R trained by SGD, but SODA trained by Adam achieves even worse accuracy than SODA trained by SGD. It shows that Adam optimizer can improve the training of data adaptors using first-order gradients but fails when using the estimated zeroth-order gradients.

Table 8: Comparing of SODA and SODA-R using SGD and Adam optimizer on CIFAR-10-C.

| Methods | SODA | SODA-R |
|---|---|---|
| SGD | 82.55 | 84.95 |
| Adam | 82.75 | 88.39 |

Table 9: Comparison of SODA and SODA-R using SGD and Adam optimizer on CIFAR-100-C.

| Methods | SODA | SODA-R |
|---|---|---|
| SGD | 52.41 | 58.32 |
| Adam | 49.75 | 60.31 |

### C.1.4 Evaluation of dropout in SODA and SODA-R

We also evaluate the effect of dropout on SODA and SODA-R. As depicted in Figure 5, a dropout layer can be inserted into the ResNet block. We conduct experiments using data adaptors with and without dropout layers for SODA and SODA-R. To keep the same network structure as SODA-R, the data adaptor used in SODA also has 2 ResNet blocks. The dropout ratio is set to be 0.5. All other settings are kept the same as in the main experiments. Table 10 and Table 11 show the results on CIFAR-10-C and CIFAR-100-C, respectively. For SODA-R, adding dropout layers can improve the accuracy by 0.7% on CIFAR-10-C and 2% on CIFAR-100-C. However, for SODA, adding dropout layers extremely hinders the performance, especially on CIFAR-100-C. This contrast indicates that dropout harms data adaptor optimized using estimated zeroth-order gradients while positively affecting data adaptor optimized using computed first-order gradients. The reason might be that the extra randomness introduced by dropout increases the difficulty of gradient estimation in zeroth-order optimization. Note that the accuracy of SODA using a data adaptor with 2 ResNet blocks on CIFAR-100-C is worse than that using a data adaptor with 0 ResNet blocks, which is consistent with the results on CIFAR-10-C as shown in Table 3.

Table 10: Comparing of SODA and SODA-R with and without dropout layers on CIFAR-10-C.

| Methods | SODA | SODA-R |
|---|---|---|
| w/ dropout | 32.19 | 88.39 |
| w/o dropout | 80.56 | 87.54 |

Table 11: Comparison of SODA and SODA-R with and without dropout layers on CIFAR-100-C.

| Methods | SODA | SODA-R |
|---|---|---|
| w/ dropout | 5.47 | 60.31 |
| w/o dropout | 43.96 | 58.27 |

To sum up, compared with SODA using zeroth-order optimization, SODA-R uses first-order optimization and adopts deeper network structure, perturbation regularization, Adam optimizer, and dropout to improve the performance. However, these techniques cannot improve or even hinder the performance of SODA. This comparison shows that the common boosting strategies used in first-order optimization cannot be directly applied to zeroth-order optimization, leading to the limited performance of methods using zeroth-order optimization.

## C.2 Convergence of SODA

In Figure 3, the convergence speeds of SODA on CIFAR-10-C and CIFAR-100-C are slower than SODA-FO and SODA-R and do not achieve complete convergence after training with 150 epochs. We further train SODA on Gaussian noise in CIFAR-10-C and CIFAR-100-C for 300 epochs to show the complete convergence of SODA as depicted in Figure 6. With more training epochs, SODA can achieve higher accuracies on both datasets. However, training with more epochs means more adaptation processing time or more computing resources with parallel computation. For time and resource efficiency, we only report the accuracies achieved at 150 epochs in our main experiments, which already improves the deployed model by a large margin. If computing time and resources are not restricted, SODA can further improve the deployed model for higher accuracy.

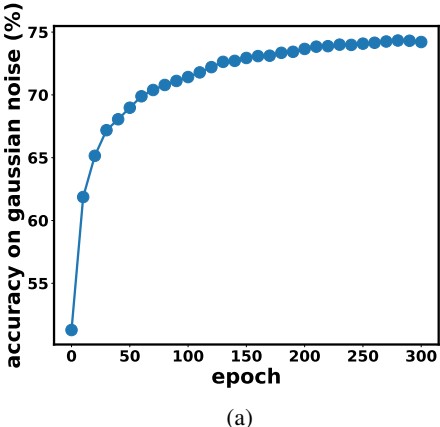

(a)

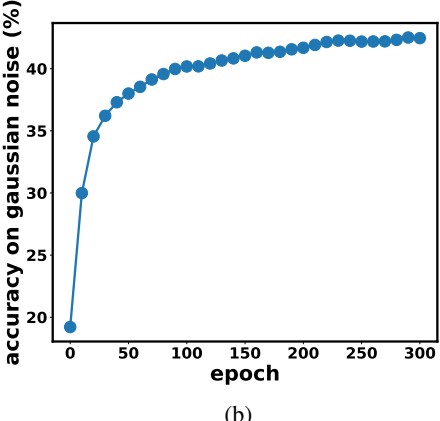

(b)

Figure 6: Accuracy convergence on Gaussian noise in (a) CIFAR-10-C and (b) CIFAR-100-C for 300 epochs.

## C.3 Hyperparameter analysis of reliable pseudo-label selection

We evaluate the hyperparameters in reliable pseudo-label selection, namely the confidence threshold $\tau$, the noise ratio $\rho$, and the balancing parameter $\alpha$. $\tau$ and $\rho$ controls the number of selected reliable pseudo-labels. With lower $\tau$ and lower $\rho$, the number of selected reliable pseudo-labels increases. We evaluate $\tau$ in $\{0.1, 0.2, 0.3, 0.4, 0.5, 0.6, 0.7, 0.8, 0.9\}$, and $\rho$ in $\{0.1, 0.3, 0.5, 0.7, 0.9\}$. Note that when $\tau = 0$, the number of selected pseudo-labels is not equal to $(1 - \rho)n$, where $n$ is the total number of test data points, because the pseudo-labels are not evenly distributed across classes as depicted in Figure 8a. The inaccurate model prediction tends to bias towards a few classes, leading to more pseudo-labels belonging to those classes. $\alpha$ controls the balance between the supervised training

objective $\mathcal{L}_{\text{ce}}$ and the unsupervised training objective $\mathcal{L}_{\text{im}}$. We evaluate $\alpha$ in $\{0.01, 0.001, 0.0001\}$. The results of SODA with different sets of hyperparameters on CIFAR-10-C Gaussian noise level 5 corruption are shown in Figure 7. The performance of SODA is stable across different hyperparameter settings. A common trend among different $\alpha$ is that accuracy tends to increase when $\tau$ and $\rho$ decrease, i.e., the top-right corner of each figure. This trend shows that the performance of the data adaptor can be improved using more selected pseudo-labels, which further indicates the reliability of the selected pseudo-labels. There is a mild tendency of performance drop in the overall performance of SODA when unsupervised learning of test points with unreliable pseudo-labels is overwhelmed by supervised learning of reliable pseudo-labels with larger $\alpha$, indicating that learning on test points with unreliable pseudo-labels also has a contribution to the performance of SODA. To balance the supervised and unsupervised learning terms, we finally choose $\alpha = 0.0001$ in our main experiments. Although better performance can be achieved by carefully fine-tuning $\tau$ and $\rho$ with a validation set to show the general performance of SODA and select the most reliable pseudo-labels for different corruptions, we set $\tau = 0.9$ and $\rho = 0.9$ for CIFAR-10/100-C tasks and $\tau = 0.1$ and $\rho = 0.9$ for ImageNet-C task in our main experiments without elaborated hyperparameter fine-tuning.

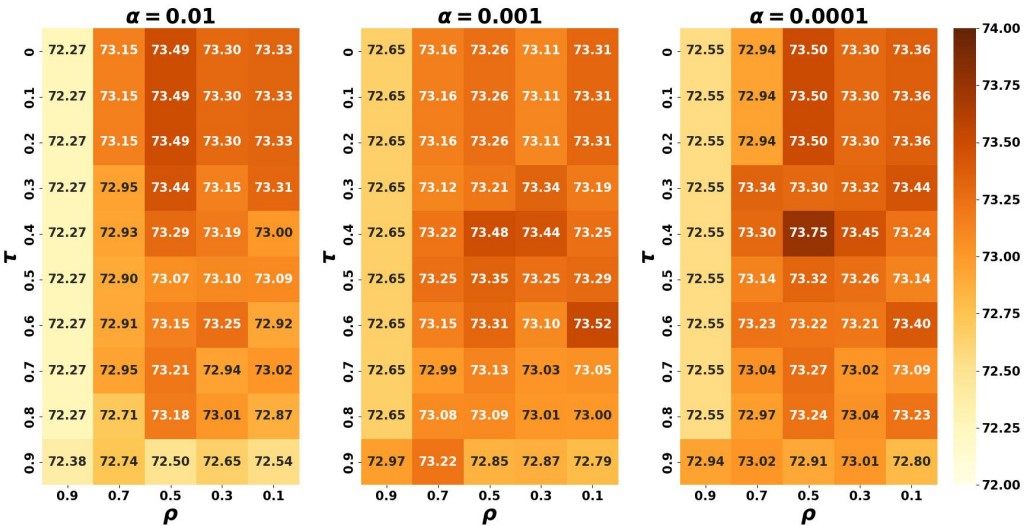

Figure 7: Evaluation of reliable pseudo-label selecting hyperparameters on CIFAR-10-C Gaussian noise corruption level 5. Numbers are prediction accuracies (%) after adaptation.

## C.4 Evaluation of queue size in SODA-O

We evaluate the effect of queue size in SODA-O. A larger queue size means more past reliable pseudo-labels and their corresponding test data points are stored and used in the adaptation process of the current mini-batch. Fixing batch size = 128, we conduct experiments on queue size $\{500, 1000, 2000, 3000\}$, and the results are shown in Table 12. The performance of SODA-O is stable across different queue sizes, especially when the queue size is smaller. When queue size increases, the ratio of reliable pseudo-labels used to train the data adaptor for the current mini-batch also increases. It makes the training of the data adaptor more biased towards the supervised training with the reliable pseudo-labels. Thus, the mild performance drop observed along with the larger queue size might indicate that the reliable pseudo-labels still have disturbance, and the unsupervised training of data points with unreliable pseudo-labels is useful to alleviate the negative effect caused by the remaining disturbance.

## C.5 Results on challenging Office-Home tasks

We conduct experiments on three challenging Office-Home tasks to show the effectiveness of SODA and SODA-R on different kinds of distribution shifts. For SODA, the network structure of the

Table 12: Comparison of SODA-O with different queue sizes. Averaged accuracies (%) over 19 corruptions are reported.

| Queue Size | 500 | 1000 | 2000 | 3000 |
|---|---|---|---|---|
| CIFAR-10-C | 78.78 | 78.79 | 78.22 | 77.73 |
| CIFAR-100-C | 47.18 | 47.21 | 46.67 | 45.63 |

data adaptor is the same as that in CIFAR-10/100-C tasks. For SODA-R, a deeper data adaptor is adopted with 2 downsampling/upsampling layers and 2 ResNet blocks. Both SODA and SODA-R are trained using SGD with learning rate = 1e-3, momentum = 0.9, and weight decay = 1e-5 for 60 epochs. A warmup process is used to first train the data adaptor with pseudo-label supervision for 10 epochs. Following DINE [22] and BETA [46], MixMatch [2] and EMA pseudo-label updating are also adopted. The weights of cross-entropy loss and mutual information loss are set to be 0.1 and 0.01, respectively. The confidence threshold $\tau$ is set to be 0.5. The noise ratio $\rho$ is set to be 0. The query number is set to 5 for SODA. The results are listed in Table 13. The results show that SODA and SODA-R can improve the deployed model in challenging Office-Home tasks with initial prediction accuracies lower than 50%

Table 13: Accuracies (%) on three challenging Office-Home domain adaption tasks. **FO Grad.** means the requirement of first-order gradient from the target model or deployed model. **Model Mod.** means the requirement of modifying the parameters of deployed models.

| Methods | FO Grad. | Model Mod. | Art->Clipart | Product->Clipart | Realworld->Clipart |
|---|---|---|---|---|---|
| Deployed | - | - | 44.67 | 40.53 | 47.12 |
| SODA | ✗ | ✗ | 46.53 | 41.47 | 48.71 |
| SODA-R | ✓ | ✗ | 50.45 | 45.20 | 53.22 |

# D  Qualitative Evaluation of SODA

## D.1  T-SNE Visualization of SODA features

To qualitatively evaluate the performance of SODA, we use T-SNE to visualize the feature embeddings of SODA, i.e., the input features of the last classification layer in the deployed model before and after adaptation in Figure 8. According to the visualization results, the feature embeddings are much more separated apart between classes after adaptation, showing the effectiveness of SODA.

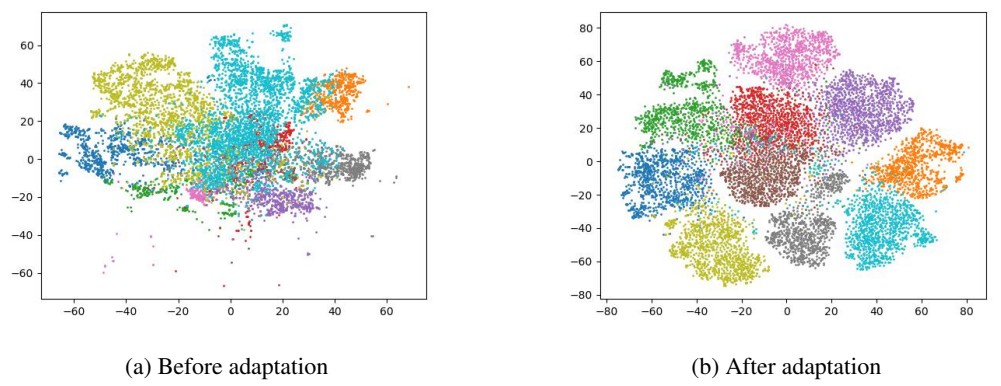

(a) Before adaptation          (b) After adaptation

Figure 8: T-SNE visualization of SODA feature embeddings on CIFAR-10-C pixelate corruption level 5.

### D.2  Examples of Adapted Data

Figure 9 shows examples of test and adapted data using SODA for 19 corruptions in CIFAR-10-C. Comparing original data without corruption, test data before adaptation, and adapted data after adaptation, it is obvious that the adapted data looks closer to the original data than the corresponding test data. This observation is consistent with the improved prediction accuracy using SODA and further illustrates that the distribution shifts between the test data and the training data are alleviated after applying SODA to test data. It also indicates that SODA adapts the test data to the deployed model by modifying them to "look like" the training data. Then, the distribution shifts between the test data and the training data are mitigated, leading to improved prediction of the deployed model.

## E  Detailed Results

There are 19 corruptions in CIFAR-10-C, CIFAR-100-C and ImageNet-C: Gaussian noise (GN), shot noise (ShN), impulse noise (IN), speckle noise (SpN), defocus blur (DB), glass blur (GlB), motion blur (MB), zoom blur (ZB), Gaussian blur (GaB), snow (SW), frost (FR), fog (FG), brightness (BR), contrast (CT), elastic transform (ET), pixelate (PX), jpeg compression (JC), spatter (SP) and saturate (SA). We report the accuracies of each method w.r.t. each corruption on CIFAR-10-C, CIFAR-100-C, and ImageNet-C in Table 14, Table 15 ,and Table 16. Except for SODA-R and MA-SO using first-order gradients from the deployed model, SODA outperforms all baselines on almost all corruptions. On CIFAR-10-C, SODA-R even outperforms MA-SO on all corruptions. On CIFAR-100-C, although the average accuracy of SODA-R is lower than that of MA-SO, SODA-R still outperforms MA-SO on 7 corruptions. On ImageNet-C, SODA improves the deployed model on most corruptions. SODA-R even outperforms or performs on par with MA-SO on 5 corruptions.

Table 14: Accuracies of 19 corruptions on CIFAR-10-C. For brevity, DA-PGD, DA-ZOO-Input, DA-PL, and DA-Direct are abbreviated as D-PG, D-Z-I, D-PL, and D-Di, respectively.

| C | Deployed | DINE | BETA | D-PG | D-Z-I | D-PL | D-Di | SODA | SODA-R | MA-SO |
|---|---|---|---|---|---|---|---|---|---|---|
| GN | 51.28 | 56.86 | 62.85 | 28.34 | 49.93 | 52.18 | 48.80 | 73.18 | 85.82 | 84.09 |
| ShN | 56.02 | 58.44 | 64.75 | 29.52 | 53.99 | 56.63 | 54.59 | 74.52 | 86.23 | 85.40 |
| IN | 42.98 | 47.25 | 53.36 | 22.23 | 42.57 | 44.38 | 41.12 | 57.48 | 83.20 | 75.06 |
| SpN | 57.15 | 59.41 | 65.61 | 27.36 | 54.33 | 57.07 | 56.04 | 73.77 | 85.52 | 84.55 |
| DB | 88.16 | 88.14 | 86.94 | 16.39 | 84.64 | 88.67 | 85.94 | 90.97 | 91.79 | 90.98 |
| GlB | 49.21 | 53.31 | 58.38 | 17.48 | 46.65 | 49.75 | 44.68 | 66.24 | 77.08 | 76.51 |
| MB | 76.62 | 77.25 | 79.27 | 17.76 | 72.05 | 77.35 | 75.16 | 86.43 | 91.13 | 87.37 |
| ZB | 89.14 | 89.37 | 88.86 | 17.76 | 85.93 | 89.73 | 87.52 | 90.96 | 92.28 | 92.46 |
| GaB | 84.59 | 84.66 | 84.65 | 15.87 | 80.24 | 85.74 | 84.38 | 91.25 | 93.13 | 90.93 |
| SW | 78.06 | 78.03 | 77.42 | 36.16 | 75.39 | 78.62 | 75.53 | 83.83 | 88.92 | 85.94 |
| FR | 71.75 | 72.39 | 72.96 | 23.15 | 68.13 | 72.24 | 70.41 | 82.92 | 87.63 | 87.45 |
| FG | 70.58 | 71.84 | 73.60 | 11.56 | 63.08 | 71.56 | 71.71 | 82.72 | 86.29 | 82.82 |
| BR | 92.98 | 92.85 | 91.28 | 41.57 | 91.46 | 92.75 | 90.18 | 92.84 | 93.15 | 92.14 |
| CT | 86.72 | 86.74 | 84.64 | 15.06 | 67.58 | 87.95 | 87.15 | 92.67 | 93.89 | 92.07 |
| ET | 76.64 | 77.35 | 78.02 | 18.32 | 72.30 | 77.04 | 71.99 | 79.50 | 82.70 | 81.99 |
| PX | 52.12 | 58.46 | 64.50 | 27.95 | 50.91 | 52.35 | 49.55 | 87.12 | 90.26 | 89.29 |
| JC | 80.55 | 80.93 | 80.70 | 29.20 | 78.51 | 81.03 | 78.60 | 86.06 | 88.08 | 87.04 |
| SP | 77.66 | 77.11 | 77.80 | 30.20 | 76.13 | 77.86 | 75.69 | 83.06 | 88.87 | 85.73 |
| SA | 93.13 | 92.90 | 92.98 | 42.09 | 91.56 | 92.68 | 90.02 | 92.94 | 93.62 | 92.43 |

Table 15: Accuracies of 19 corruptions on CIFAR-100-C. For brevity, DA-PGD, DA-ZOO-Input, DA-PL, and DA-Direct are abbreviated as D-PG, D-Z-I, D-PL, and D-Di, respectively.

| C | Deployed | DINE | BETA | D-PG | D-Z-I | D-PL | D-Di | SODA | SODA-R | MA-SO |
|---|---|---|---|---|---|---|---|---|---|---|
| GN | 19.21 | 20.17 | 20.89 | 5.31 | 13.95 | 19.13 | 16.73 | 40.88 | 53.59 | 57.25 |
| ShN | 22.13 | 23.02 | 24.23 | 5.28 | 16.20 | 21.87 | 19.49 | 41.91 | 55.16 | 58.60 |
| IN | 12.26 | 11.50 | 11.78 | 3.70 | 9.28 | 12.38 | 10.32 | 20.31 | 49.14 | 47.37 |
| SpN | 23.37 | 23.84 | 25.02 | 4.69 | 16.64 | 23.27 | 20.39 | 40.35 | 53.72 | 58.42 |
| DB | 60.39 | 57.75 | 56.31 | 3.24 | 44.99 | 60.00 | 55.37 | 67.22 | 68.03 | 68.92 |
| GlB | 17.74 | 17.81 | 18.58 | 3.01 | 12.61 | 17.22 | 12.90 | 29.92 | 41.97 | 49.66 |
| MB | 45.79 | 43.98 | 43.20 | 3.61 | 34.81 | 46.38 | 43.43 | 58.50 | 66.88 | 64.03 |
| ZB | 61.64 | 59.07 | 57.06 | 3.42 | 44.73 | 61.98 | 57.18 | 64.86 | 66.57 | 70.51 |
| GaB | 54.40 | 51.94 | 49.67 | 3.20 | 39.83 | 55.09 | 51.40 | 68.53 | 70.47 | 69.53 |
| SW | 45.47 | 44.82 | 43.18 | 6.08 | 35.14 | 44.88 | 40.18 | 50.30 | 59.15 | 58.23 |
| FR | 39.77 | 39.65 | 39.52 | 3.49 | 27.85 | 39.77 | 37.11 | 52.10 | 57.82 | 60.97 |
| FG | 31.94 | 31.23 | 30.71 | 1.43 | 17.73 | 31.66 | 31.07 | 48.71 | 55.13 | 54.40 |
| BR | 71.18 | 69.67 | 65.47 | 6.53 | 61.33 | 70.23 | 64.35 | 70.29 | 70.73 | 70.34 |
| CT | 49.10 | 46.10 | 43.35 | 1.23 | 25.49 | 51.56 | 48.43 | 71.71 | 73.03 | 69.91 |
| ET | 40.45 | 39.86 | 38.89 | 3.29 | 28.36 | 39.66 | 32.71 | 40.14 | 49.77 | 56.61 |
| PX | 27.77 | 27.87 | 29.36 | 4.24 | 29.23 | 27.72 | 24.85 | 55.63 | 62.06 | 66.37 |
| JC | 49.98 | 49.50 | 48.39 | 5.91 | 44.13 | 50.36 | 45.79 | 55.99 | 59.92 | 63.89 |
| SP | 44.18 | 43.80 | 42.47 | 5.03 | 35.22 | 44.56 | 39.20 | 49.49 | 61.85 | 61.83 |
| SA | 69.97 | 68.25 | 64.69 | 6.24 | 61.55 | 69.64 | 64.36 | 69.73 | 70.82 | 71.48 |

Table 16: Accuracies of 19 corruptions on ImageNet-C. For brevity, DA-PGD, DA-ZOO-Input, DA-PL, and DA-Direct are abbreviated as D-PG, D-Z-I, D-PL, and D-Di, respectively. Note that DINE and BETA are excluded because their required initialization of the target models violates the setting of test-time adaptation.

| C | Deployed | D-PG | D-Z-I | D-PL | D-Di | SODA | SODA-R | MA-SO |
|---|---|---|---|---|---|---|---|---|
| GN | 8.98 | 6.72 | 10.52 | 10.38 | 9.18 | 26.72 | 40.99 | 34.85 |
| ShN | 11.48 | 9.39 | 6.14 | 12.93 | 9.97 | 28.46 | 43.24 | 40.46 |
| IN | 8.77 | 8.30 | 11.47 | 10.38 | 8.51 | 25.31 | 56.44 | 35.60 |
| SpN | 29.27 | 21.82 | 19.35 | 32.23 | 30.05 | 46.62 | 56.26 | 54.18 |
| DB | 21.99 | 1.29 | 2.42 | 21.73 | 17.47 | 24.52 | 31.43 | 43.98 |
| GlB | 7.35 | 2.30 | 0.74 | 6.78 | 1.69 | 7.84 | 0.62 | 38.46 |
| MB | 20.67 | 3.20 | 3.55 | 19.07 | 15.44 | 31.72 | 58.76 | 63.46 |
| ZB | 32.81 | 7.11 | 16.42 | 30.94 | 26.08 | 30.64 | 43.74 | 67.36 |
| GaB | 17.88 | 1.02 | 2.03 | 17.07 | 12.86 | 15.38 | 0.69 | 33.93 |
| SW | 34.92 | 1.33 | 14.55 | 33.98 | 31.15 | 51.58 | 59.39 | 63.52 |
| FR | 36.88 | 1.63 | 7.57 | 37.70 | 36.27 | 47.00 | 54.42 | 58.40 |
| FG | 57.72 | 0.91 | 32.85 | 58.07 | 57.87 | 60.92 | 60.45 | 73.17 |
| BR | 73.11 | 45.92 | 60.78 | 74.11 | 72.22 | 73.04 | 60.91 | 75.02 |
| CT | 70.92 | 0.13 | 50.83 | 75.02 | 75.18 | 74.72 | 61.71 | 69.21 |
| ET | 9.88 | 12.31 | 1.84 | 8.40 | 5.82 | 28.43 | 53.61 | 65.50 |
| PX | 7.29 | 22.29 | 3.38 | 6.74 | 4.49 | 53.99 | 58.25 | 66.37 |
| JC | 47.32 | 43.11 | 24.12 | 49.15 | 47.63 | 51.08 | 52.52 | 59.22 |
| SP | 33.03 | 19.70 | 14.93 | 34.61 | 30.26 | 54.21 | 66.05 | 65.92 |
| SA | 65.53 | 42.17 | 50.42 | 67.06 | 65.87 | 69.25 | 65.87 | 72.50 |

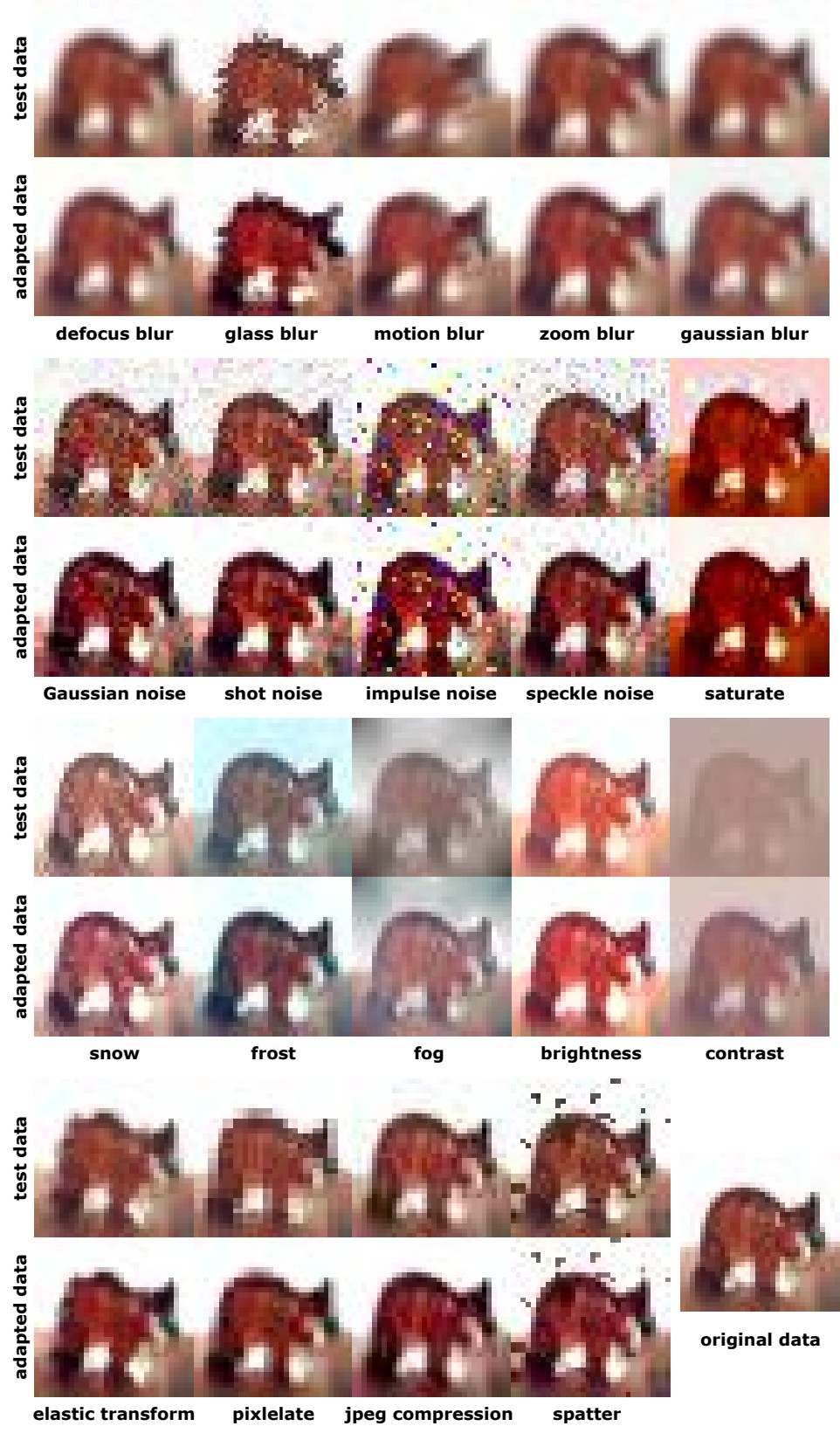

Figure 9: Examples of test data and adapted data using SODA for 19 corruptions in CIFAR-10-C. The bottom-right data is the original data in the CIFAR-10 test dataset without corruption.

