# OpenReview forum: "SODA: Robust Training of Test-Time Data Adaptors"
_NeurIPS.cc/2023/Conference — NeurIPS 2023 poster_

### Official Review · Reviewer_H74J · 2023-07-04

**Soundness:** 3 good
**Presentation:** 3 good
**Contribution:** 3 good
**Rating:** 6
**Confidence:** 3

**Summary:**

This paper proposes SODA, a test-time data adaptor with the black-box source model leveraging Zeroth-Order Optimization for the adaptor, which involves a random perturbation on the adaptor model's parameters. It also considers the scenario, namely SODA-R when the gradient information is available and online setting SODA-O. The proposed method outperforms existing benchmarks, including BETA and DINE. The experiment is comprehensive with CIFAR10-C/CIFAR100-C. However, having more benchmark dataset experiments is better for achieving better confidence with the proposed framework for practical application.

**Strengths:**

Test-time adaptor problem is well-motivated, and leveraging the adaptor could potentially save the training cost in practice.
Under the limited available information scenario, the authors claim that they successfully tackle the problem better than other benchmarks, such as BETA and DINE. I value the simplicity of the approach only with two components; 1) mutual information maximization and 2) cross-entropy loss with pseudo labels by achieving a better performance. As a black-box scenario, overcoming the lack of gradient is another main ingredient of this paper, and it outperforms other benchmarks in CIFAR10-C/CIFAR100-C. Application of Zeroth-Order Optimization is another key ingredient, but the simplicity is better appealing to me.

**Weaknesses:**

1. I would like to understand the relationship between the model perturbation and the data augmentation in the proposed framework. See the more detailed question in the Questions section.
2. A good combination of the choice $\sigma, \alpha, \tau$ seems to be critical. I see this discussion at C3 in the Appendix, but I suggest that the authors make this more explicit in the main paper. If those parameters are dataset/or task-dependent, it's not yet applicable in practice.
3. Why $\sigma=0.5$ generally achieve a better performance? And a better performance with $\alpha=0.0001$ implies that the mutual information term is the most critical. A further ablation study of each loss term would be necessary.

Minor Comments:
- L61: SOTA $\to$ SODA?

**Questions:**

1. Do we need to apply data augmentations in the SODA framework? I guess not, but data augmentation information, including DINE and BETA, seems missing. In another sense, data augmentation conflicts (like Dropouts) with applying the model perturbation in mutual information calculation - Eq. $(7)$. So if we don't have to apply Data Augmentation, what is the primary source of the outperformance? I would like to see some convincing evidence of the hypothesis.
2. Aligning with the first question, what is the source of the randomness in calculating Eq. $(7)$? Specifically, the perturbation on $\theta$? or data $x_i$? I would like to have confirmation from the authors.

**Limitations:**

The main limitation of this paper is the scope of the experiments, although an extensive study in CIFAR10-C/CIFAR100-C. There are many other benchmarks in the domain-adaptation, such as Office-31, VISDA, Office-Home, etc. It would be more convincing if the authors could demonstrate the out-performance with a fixed set of hyperparameters.

---

> ### Author Rebuttal · Authors · 2023-08-09
>
> > **W1:** I would like to understand the relationship between the model perturbation and the data augmentation in the proposed framework.
>
> **AW1:** There are **two kinds of “perturbation”** in our proposed work:
> - **Parameter perturbation** used in gradient estimation of zeroth-order optimization (ZOO) [1]. It is used to overcome the inaccessible gradient problem.
> - **Data adaptation** achieved by the data adaptor.
>
> **Data adaptation is different from data augmentation** in two points:
>
> - Working scheme: data adaptation generates perturbations using the network and adds them to the original data samples, while data augmentation performs pre-defined visual transformations to data samples.
> - Purpose: the perturbations generated in data adaptation are used to adapt test data to the deployed model, while data augmentation is usually to reduce overfitting in neural network training.
>
> > **Q1:** Do we need to apply data augmentations in the SODA framework? I guess not, but data augmentation information, including DINE and BETA, seems missing.
>
> **AQ1:** Data augmentation is not considered in SODA and DINE, but is used in BETA.
>
> > **Q2:** Aligning with the first question, what is the source of the randomness in calculating Eq. (7)? Specifically, the perturbation on $\theta$ or data $x_i$? I would like to have confirmation from the authors.
> >
> > **Q3:** In another sense, data augmentation conflicts (like Dropouts) with applying the model perturbation in mutual information calculation - Eq. (7).
>
> **AQ2&3:** First, we would like to explain the working scheme of mutual information maximization (IM). IM [2] has two terms:
> - Conditional entropy which encourages the model prediction to be more certain and form tighter clusters for each class.
> - Marginal entropy which encourages the model prediction to be more diverse and form more separated clusters among classes.
>
> In the calculation of IM, the randomness comes from the distribution of the test data and the uncertainty of the model prediction, not from perturbations on either $\boldsymbol{\theta}$ or $\mathbf{x}_i$. Data augmentation is not conflicted with the calculation of IM in Eq.(7).
>
> > **Q4:** So if we don't have to apply Data Augmentation, what is the primary source of the outperformance? I would like to see some convincing evidence of the hypothesis.
>
> **AQ4:** We agree that data augmentation is a powerful method to improve the performance of models, but the key idea of SODA is data adaptation instead of data augmentation. The effectiveness of SODA comes from the training of the data adaptor to generate adapted data. The training objective consists of two components:
> - Supervised training with reliable pseudo-labels to alleviate the data corruption problem caused by unreliable pseudo-labels.
> - Unsupervised training of data samples with unreliable pseudo-labels to encourage the model prediction on those data samples to be certain and diverse.
>
> > **W2:** A good combination of the choice $\sigma$, $\alpha$, $\tau$ seems to be critical. I see this discussion at C3 in the Appendix, but I suggest that the authors make this more explicit in the main paper.
>
> **AW2:** Thanks for your constructive suggestions. We agree with the point that choosing a good combination of hyper-parameters may be suboptimal. But as discussed in Appendix C3, our proposed SODA is robust to most combinations. With your kind reminder, we will put this discussion on our main page.
>
> > **W3:** If those parameters are dataset/or task-dependent, it's not yet applicable in practice.
>
> **AW3:** Thanks for your insightful comments. We also agree that the need of choosing good hyper-parameters may hinder the practicability of our work. But for CIFAR-10-C and CIFAR-100-C datasets used in our main experiments, accuracies before adaptation range from ~10% to ~90% for different corruptions. Using a fixed threshold and ratio, our proposed SODA improves the deployed model for almost all corruptions, showing that SODA can handle distribution shifts to various extents, and is not specifically dataset dependent.
>
> > **W4:** Why $\sigma$ = 0.5 generally achieve a better performance?
>
> **AW4:** The hyper-parameter analysis experiments are conducted on CIFAR-10-C Gaussian noise corruption. Before adaptation, the initial accuracy is 51.28%. Since $\rho$ ($\sigma$ in your question) should ideally be the noise ratio in the pseudo-labels, i.e. the error rate of the deployed model before adaptation, setting $\rho=0.5$ is expected to have better results. However, in our main experiments, we do not elaborately choose different $\rho$ for different kinds of corruption with different initial accuracies. SODA with a fixed $\rho$ has already improved the deployed model to a large extent.
>
> > **W5:** And a better performance with $\alpha$ = 0.0001 implies that the mutual information term is the most critical. A further ablation study of each loss term would be necessary.
>
> **AW5:** Thanks for your instructive suggestion. Both loss terms in SODA are useful. Besides the discussion in Appendix C3, we also show the effectiveness of each loss term in the Office-Home Art->Clipart task:
>
> |Office-Home|Deployed|Pseudo-label only|IM only|SODA|
> |-|-|-|-|-|
> |Art->Clipart|44.47%|44.99%|45.36%|46.53%
>
> > **Minor Comments**: L61: SOTA  SODA?
>
> **A:** Thanks for pointing out this typo, it should be SODA, we will fix it in our revised paper.
>
> **Answer for limitations:** With your constructive suggestion, we provide more experimental results in the general response. We also agree that using adaptive thresholds might be a promising improvement in the future.
>
> > References:
> >
> > [1] Sijia Liu, et al. A primer on zeroth-order optimization in signal processing and machine learning: Principals, recent advances, and applications. IEEE Signal Processing Magazine, 2020.
> >
> > [2] Shi, Yuan, and Fei Sha. Information-theoretical learning of discriminative clusters for unsupervised domain adaptation. ICML 2012.

---

> > ### Comment · Reviewer_H74J · 2023-08-12
> >
> > I appreciate the authors taking the time to answer all questions during the rebuttal. After carefully checking all answers and the source codes, I'm more convinced of the results. Although the real-world scenario deployment is still in question, the adapter combined with the mutual information setting seems to work well under synthetic noise scenarios and is worth presenting. Therefore, I raise my rating one step more.

---

> > > ### Author Response · Authors · 2023-08-13
> > > **Thanks to Reviewer H74J!**
> > >
> > > We are glad to hear that our response has addressed your questions. Thanks for upgrading your score!

---

### Official Review · Reviewer_xmoq · 2023-07-06

**Soundness:** 2 fair
**Presentation:** 2 fair
**Contribution:** 2 fair
**Rating:** 5
**Confidence:** 3

**Summary:**

In this work, the authors aim to adapt unlabelled test data to a deployed model without access to its parameters and inner structures during the testing process. Specifically, the authors utilize a data adaptor during testing to map test data into the deployed model, which gradients are estimated via ZOO. Experiments are conducted on CIFAR-10C and CIFAR-100C to verify the effectiveness of the proposed method.

**Strengths:**

1. This paper is well-written and easy-to-follow.

2. The code is available with the submission which improves the reproducibility index of the paper.


**Weaknesses:**

1. Experiments are not convincing. The authors only use CIFAR-10C and CIFAR-100C to verify the effectiveness of their algorithm. I suggest more datasets with various types of distribution shift should be included such as Office-31, Office-Home, PAC, etc.

2. The necessity of ZOO is not clear. I can understand parameters of the deployed model are inaccessible, but why the data adaptor you generate during testing is still a black box? If the parameters of the data adaptor are known, why do not you simply fix the parameters of the deployed model as constants, and use FOO to calculate gradients of the data adaptor?

3. More baselines should be considered. There already exist some TTA algorithms that do not requiring access to parameters of the deployed model, such as T3A [1].

[1] Iwasawa Y, Matsuo Y. Test-time classifier adjustment module for model-agnostic domain generalization[J]. Advances in Neural Information Processing Systems, 2021, 34: 2427-2440.

**Questions:**

See the Weaknesses.

**Limitations:**

The authors provide the limitations of their work.

---

> ### Author Rebuttal · Authors · 2023-08-09
>
> > **Q1:** Experiments are not convincing. The authors only use CIFAR-10C and CIFAR-100C to verify the effectiveness of their algorithm. I suggest more datasets with various types of distribution shift should be included such as Office-31, Office-Home, PAC, etc.
>
> **A1:** Thanks for your instructive suggestion, we further conduct experiments on ImageNet-C and challenging Office-Home domain adaptation tasks to illustrate the efficacy of our proposed SODA framework. The results are shown in the attached file in the general response.
>
> > **Q2:** The necessity of ZOO is not clear. I can understand parameters of the deployed model are inaccessible, but why the data adaptor you generate during testing is still a black box? If the parameters of the data adaptor are known, why do not you simply fix the parameters of the deployed model as constants, and use FOO to calculate gradients of the data adaptor?
>
> **A2:** Sorry about the confusion. In our work, the data adaptor is not a black box, and its parameters can be accessed and modified. In our settings, the parameters of the deployed model are hidden, so **backward propagation through the deployed model is not allowed**, leading to infeasible gradient computation. Hence, zeroth-order optimization is used to circumvent this problem and estimate gradients w.r.t. parameters of the data adaptor for training of the data adaptor.
>
> > **Q3:** More baselines should be considered. There already exist some TTA algorithms that do not require access to parameters of the deployed model, such as T3A [1].
>
> **A3:** Thanks for your constructive suggestion. We also expect more proposed works dealing with the same settings as we do, however, most of the existing works which do not modify the model parameters require access to the extracted features. For example, T3A[1] splits the pre-trained model into a feature extractor and classifier, and uses the features extracted from the feature extractor to form the support set. Compared to their settings, our settings forbid access to features that are stricter and more practical due to intellectual property protection, misuse prevention, privacy concerns in healthcare and finance, etc.
>
> > References:
> >
> > [1] Iwasawa Y, Matsuo Y. Test-time classifier adjustment module for model-agnostic domain generalization[J]. Advances in Neural Information Processing Systems, 2021, 34: 2427-2440.

---

> > ### Comment · Reviewer_xmoq · 2023-08-13
> >
> > Dear Authors,
> >
> > Thank you so much for carefully considering the comments in my review. My concerns have been addressed. I also notice that the authors provide a theoretical analysis. Therefore, I have raised my score.
> >
> > Best regards,
> >
> > Reviewer xmoq

---

> > > ### Author Response · Authors · 2023-08-13
> > > **Thanks to Reviewer xmoq**
> > >
> > > Dear Reviewer xmoq,
> > >
> > > We are glad to hear that our response has addressed your concerns. Thanks for raising your score!
> > >
> > > Best regards,
> > >
> > > Authors of #4388

---

### Official Review · Reviewer_SVqY · 2023-07-10

**Soundness:** 3 good
**Presentation:** 3 good
**Contribution:** 2 fair
**Rating:** 5
**Confidence:** 4

**Summary:**

This paper proposes usage of zeroth order optimization (ZOO) for test-time adaptation (TTA) to ease several practical issues regarding accessing model parameters during TTA. Since the ZOO with pseudo label, which is a standard method in TTA, might cause the unreliable gradient, the paper proposes a sample selection method using the confidence of the prediction and class balance, and use only reliable sample to compute pseudo-label loss. The unreliable sample is used to compute another unsupervised loss to facilitate better adaptation. They show its effectiveness, CIFAR10-C and CIFAR100-C. No theory is provided.

**Strengths:**

1. The paper is generally well written and easy to follow.
2. The usage of zeroth-order optimization in TTA is well motivated and interesting new problem.
3. The proposed sample selection approach based on the confidence and class balance seems not to be revolutionary but sensible.

**Weaknesses:**

1. The experiment is limited to the CIFAR10-C and CIFAR100-C. As usual, I recommend adding experiments on ImageNet-C and some domain adaptation datasets.

2. The technical novelty is not high. Besides, I found that the effectiveness of the selection method proposed in this paper is not fully validated. For example, the paper does not provide ablation about the sensitivity about the threshold parameter. I'm also curious why we should not compute information maximization loss for reliable samples. Besides, the necessity of information maximization loss is not experimentally validated.

3. The selection of the information maximization loss is not well described. Why did you choose the specific loss function?

4. Several details are unclear for me. Including,

- What the difference between SODA-R and SODA-FO?
- I'm confused by the table 5, since it says that SODA-O as a variant of SODA under *online* settings but seems to repeat the optimization multiple epochs. Do you repeat the optimization after you reach all test dataset? In the case, I have to say that it is not usual online setup.

5. No theoretical results.

**Questions:**

See weakness section

**Limitations:**

None.

---

> ### Author Rebuttal · Authors · 2023-08-09
>
> > **Q1:** The experiment is limited to the CIFAR10-C and CIFAR100-C. As usual, I recommend adding experiments on ImageNet-C and some domain adaptation datasets.
>
> **A1:** Thanks for your constructive suggestions. More experimental results on ImageNet-C and challenging Office-Home tasks are shown in the attached file in general response.
>
> > **Q2:** The technical novelty is not high.
>
> **A2:** We would like to highlight our main novelty and contributions in the following three folds:
> - **ZOO for test-time data adaptation**: We tackle a challenging and realistic setting where the parameters of the deployed model are inaccessible and unmodifiable. zeroth-order optimization and data adaptation are proposed to solve this problem.
> - **Label noise in ZOO**: We analyze the effect of label noise in ZOO, and point out that the noisy pseudo-labels can cause biased gradient estimation in ZOO, leading to limited performance of test-time data adaptor.
> - **New methods for robust test-time data adaptation**: Based on our analysis, we propose SODA to robustly train the test-time data adaptor. SODA separates the pseudo-labels into reliable and unreliable sets and performs semi-supervised learning using cross-entropy loss and mutual information maximization.
>
> > **Q3:** Besides, I found that the effectiveness of the selection method proposed in this paper is not fully validated. For example, the paper does not provide ablation about the sensitivity about the threshold parameter.
>
> **A3:** The hyper-parameter analysis is presented in Appendix C.3. In response to your kind reminder, we will highlight it in the main paper.
>
> > **Q4:** I'm also curious why we should not compute information maximization loss for reliable samples.
>
> **A4:** Information maximization works by making the model prediction more certain while keeping diversity in the global structure. The predictions of data samples with reliable labels already have high confidence, thus information maximization is not needed for those samples.
>
> > **Q5:** Besides, the necessity of information maximization loss is not experimentally validated.
>
> **A5:** The baseline DA-PL in our main experiments shows the effectiveness of information maximization. DA-PL only uses the pseudo-labels to train the data adaptor and makes trivial improvements. Compared to DA-PL, our proposed SODA improves the deployed model to a large extent by separating the dataset into a reliable set supervised trained by reliable pseudo-labels and an unreliable set unsupervised trained by information maximization.
>
> > **Q6:** The selection of the information maximization loss is not well described. Why did you choose the specific loss function?
>
> **A6:** Thanks for your helpful comments. Because of the high error rate, data samples with unreliable pseudo-labels may be misclassified into classes with large amounts of samples, and hard to separate them. Following previous works [1][2], One useful way to circumvent this problem is to encourage diversity among predictions of each data sample. Information maximization is a widely-used unsupervised loss that can encourage both global diversity and local certainty of model predictions. In response to your kind reminder, we will add this explanation to our revised paper.
>
> > **Q7:** Several details are unclear for me. Including, What the difference between SODA-R and SODA-FO
>
> **A7:** SODA-R and SODA-FO are both relaxed baselines assuming that gradient computation is allowed from the deployed model while the parameters of the deployed model are still not modifiable. They both use first-order optimization to compute gradients for the training of the data adaptor. As a comparison baseline, SODA-FO keeps everything the same as SODA except the usage of FOO, to show the effect of ZOO in the training of the data adaptor. Based on SODA-FO, SODA-R adopts deeper network architecture of the data adaptor, Adam optimizer, perturbation regularization, and dropout strategy to show better results which can be achieved by SODA under relaxed settings. A more detailed discussion is presented in Appendix C.1. Thanks for your comments, we will put a more detailed explanation in our revised paper.
>
> > **Q8:** I'm confused by the table 5, since it says that SODA-O as a variant of SODA under online settings but seems to repeat the optimization multiple epochs. Do you repeat the optimization after you reach all test dataset? In the case, I have to say that it is not usual online setup.
>
> **A8:** Sorry about the confusion. The optimization in SODA-O is not repeated after reaching the entire test dataset but only repeats for the current test data batch and the cached queue. During the adaptation of the current test data batch, the previous data batches are no longer available except for those saved in the queue. After reaching the entire test dataset, the whole adaptation process ends. Thanks for your comments, we will put a more detailed explanation in our revised paper.
>
> > **Q9:** No theoretical results.
>
> **A9:** Thanks for your instructive comments. We provide theoretical analysis about pseudo-label-robust training and ZOO in the general response.
>
> > References:
> >
> > [1] Jian Liang, et al. Do we really need to access the source data? source hypothesis transfer for unsupervised domain adaptation. ICML, 2020.
> >
> > [2] Jian Liang, et al. Dine: Domain adaptation from single and multiple black-box predictors. CVPR, 2022

---

> ### Author Response · Authors · 2023-08-18
> **Please check if our response clarified your questions**
>
> Dear Reviewer SVqY,
>
> As the discussion period ends soon, we just wanted to check if our response clarified your questions. Thanks again for your constructive feedback.
>
> Best regards,
>
> Authors of#4388

---

> > ### Comment · Reviewer_SVqY · 2023-08-21
> >
> > Thank you for providing detailed responses. I think the response resolve my initial concerns to a good extent. I therefore increase my score and slightly leaning toward acceptance of the paper.

---

> > > ### Author Response · Authors · 2023-08-21
> > > **Thanks to Reviewer SVqY**
> > >
> > > Dear Reviewer SVqY,
> > >
> > > We are glad to hear that our responses have resolved your concerns. Thank you for increasing your score!
> > >
> > > Best regards,
> > >
> > > Authors of #4388

---

### Official Review · Reviewer_1NX6 · 2023-07-11

**Soundness:** 3 good
**Presentation:** 3 good
**Contribution:** 3 good
**Rating:** 5
**Confidence:** 3

**Summary:**

To better adapt models to test distributions without changing model parameters, this paper utilizes the strategy that trains a data adaptor which can adjust the test data to fit the deployed models. To avoid the potential corruption of data features caused by the data adaptor, the proposed method treats the test-time adaptation process as a semi-supervised learning process. Specifically, the test data points are split into two subsets, including a high confidence set to perform regular cross entropy minimization and a low confidence set (as unlabeled set) to perform mutual information maximization.

**Strengths:**

1. This paper studies a realistic problem in test time adaptation, i.e., the unreliable nature of the pseudo-labels assigned to the test data.

2. This paper proposes solves the low-quality issue of pseudo-labels by transforming the adaptation process as a semi-supervised learning process, in which the data adaptor model is less impacted by the mislabeled data points.

3. The proposed method uses ZOO framework to estimate the gradients of the parameters and can efficiently the problem with a few queries.


**Weaknesses:**

1. The Pseudo-Label-Robust Data Adaptation module is the key contribution of this paper, but the design of this part is too simple. The problem here is actually noisy label learning problem and treating it as semi-supervised learning is a common strategy.

2. The reliable pseudo-label selection process in Subsection 3.3 utilizes fixed threshold or ratio to select data points, which maybe not robust in real-world settings.

3. The experiments on large scale datasets are not presented in the paper. Moreover, the parameter analyses in terms of $\tau$ and $\rho$ are needed.

**Questions:**

Please refer to Weaknesses.

**Limitations:**

N/A.

---

> ### Author Rebuttal · Authors · 2023-08-09
>
> > **Q1**: The Pseudo-Label-Robust Data Adaptation module is the key contribution of this paper, but the design of this part is too simple. The problem here is actually noisy label learning problem and treating it as semi-supervised learning is a common strategy.
>
> **A1**: We agree with your point that Pseudo-Label-Robust Data Adaptation is one of the key contributions, but we would like to highlight that the **challenges** have three folds.
>
> - **Inaccessible model parameters**. The parameters of target models are inaccessible, thus existing model adaptation methods may fail to promote the performance of target models. Accordingly, we propose to employ a data adaptor for target models.
> - **Infeasible gradients**. Calculating gradients using target models no longer holds in the scenario since model parameters are hidden. Therefore, we propose to employ zero-order optimization (ZOO) to approximate gradients for the update of the data adaptor.
> - **Noisy label**. The label of test samples is unknown, leading to biased loss values used in ZOO. To this end, we employ the commonly used pseudo-label strategy to perform robust data adaption.
>
> Moreover, inspired by your constructive comments, we further give **theoretical analysis** of the mentioned pseudo-label strategy under test-time adaptation scenarios, as shown in the general response.
>
> > **Q2:** The reliable pseudo-label selection process in Subsection 3.3 utilizes fixed threshold or ratio to select data points, which maybe not robust in real-world settings.
>
> **A2:** Thanks for your insightful comments. We agree with the point that fixed threshold and ratio may be suboptimal. As discussed in Appendix C.3, a good combination of threshold and ratio may be able to achieve better results. Adaptive noise ratio estimation like Gaussian Mixture Model (GMM)[1] might be of help to make SODA more robust. Meanwhile, we can see that SODA still outperforms baselines using fixed threshold and ratio.
>
> > **Q3:** The experiments on large scale datasets are not presented in the paper.
>
> **A3:** Thanks for your constructive suggestion, which motivates us to verify the efficacy of SODA on a large-scale dataset, i.e., ImageNet-C. The results are shown in the general response. We will add these results and discussion to our revised paper.
>
>
> > **Q4:** Moreover, the parameter analyses in terms of $\tau$ and $\rho$ are needed.
>
> **A4:** The hyper-parameter analyses are presented in Appendix C.3. In response to your kind reminder, we will highlight them on the main page.
>
> > References:
> >
> > [1] E. Arazo, et al, Unsupervised label noise modeling and loss correction, ICML, 2019.

---

> > ### Comment · Reviewer_1NX6 · 2023-08-22
> > **Thanks for the responses**
> >
> > Your responses have addressed some of the previous concerns. Nevertheless, I still retain concern about the hyperparameter study. Presently, the hyperparameter analysis is confined to CIFAR-10. However, it would be valuable to extend this study to diverse datasets to figure out whether hyperparameters, such as the threshold $\tau$, exhibit substantial variation across different datasts. Consequently, I will keep my current evaluation score.

---

### Author Rebuttal · Authors · 2023-08-09

We sincerely appreciate all reviewers for taking the time and effort to review our paper and provide valuable feedback. We would like to thank reviewers for their recognition of our work: 1) our problem is **realistic** (#1NX6), **well-motivated** and **interesting new** (#SVqY and #H74J); 2) our method is **efficient** (#1NX6), **effective** (#SVqY, #xmoq, #H74J), **simple** (#H74J) and **sensible** (#SVqY); 3) our paper is **well-written** and **easy-to-follow** (#SVqY and #xmoq).

Besides the response to each reviewer, we would like to provide more **experimental results** on ImageNet-C and challenging Office-Home domain adaptation tasks as constructively suggested by reviewers in the attached PDF file.

Furthermore, we would like to provide more **theoretical analysis** here to show that our proposed pseudo-label-robust training strategy can tighten the upper bound of the expected gradient estimation error in zeroth-order optimization.
>
> Given a data adaptor $\mathbf{G}$ with parameter $\boldsymbol{\theta}$, a deployed model $\mathbf{M}$ and a test dataset $\mathbf{X} = \\{\mathbf{x}_1,...,\mathbf{x}_n\\}$, denote the adapted data sample as ${\mathbf{x}_i^{\boldsymbol{\theta}}}$, the true label of $\mathbf{x}_i$ as $\mathbf{y}_i$, and $\hat{\mathbf{p}}_i^{\boldsymbol{\theta}}=\mathbf{M}\circ \mathbf{G}(\mathbf{x}_i;\boldsymbol{\theta})$.
>
> According to [1], minimizing the cross entropy loss $\mathcal{L}_{\rm ce}(\mathbf{y}_i, \hat{\mathbf{p}}_i^{\boldsymbol{\theta}})$ is equivalent to maximizing the mutual information $\mathcal{L} _{\rm im}(\mathbf{x}_i^{\boldsymbol{\theta}})$.
>
> From derivation in Appendix A, with pseudo-label $\hat{\mathbf{y}}_i = \mathbf{y}_i + \boldsymbol{\sigma}_i$, the KL divergence loss at test data point $\mathbf{x}_i$ is:
> $$\mathcal{L}_i = -H(\mathbf{y}_i+\boldsymbol{\sigma}_i)+\mathcal{L} _{\rm ce}(\mathbf{y}_i, \hat{\mathbf{p}}_i^{\boldsymbol{\theta}}) - \boldsymbol{\sigma}_i \log \hat{\mathbf{p}}_i^{\boldsymbol{\theta}}.$$
>
> Denoting $h(\mathbf{x}_i) = -\boldsymbol{\sigma}_i \log \hat{\mathbf{p}}_i^{\boldsymbol{\theta}}$, the gradient of the KL divergence loss is:
> $$\nabla _{\boldsymbol{\theta}}\mathcal{L}_i = \nabla _{\boldsymbol{\theta}}\mathcal{L} _{\rm ce} + \nabla _{\boldsymbol{\theta}}h.$$
>
> Then, in gradient estimation of ZOO, the estimated gradient of the KL divergence loss is:
> $$\widehat{\nabla} _{\boldsymbol{\theta}}{\check{\mathcal{L}} _i} = \widehat{\nabla} _{\boldsymbol{\theta}}\mathcal{L} _{\rm ce} + \widehat{\nabla} _{\boldsymbol{\theta}}h.$$
>
> Hence, before applying pseudo-label-robust data adaptation, the upper bound of expected gradient estimation error is:
> $$\mathbb{E}[\parallel \widehat{\nabla} _{\boldsymbol{\theta}}{\check{\mathcal{L}} _i} - \nabla _{\boldsymbol{\theta}}\mathcal{L} _i \parallel_2] \leq \mathbb{E}[\parallel \widehat{\nabla} _{\boldsymbol{\theta}}{\check{\mathcal{L}} _{\rm ce}} - \nabla _{\boldsymbol{\theta}}\mathcal{L} _{\rm ce} \parallel_2] + \mathbb{E}[\parallel \widehat{\nabla} _{\boldsymbol{\theta}}h - \nabla _{\boldsymbol{\theta}}h \parallel_2].$$
>
> In SODA, by separating $\mathbf{X}$ into reliable set $\mathbf{X}_r$ learned by cross-entropy loss with pseudo-labels, and unreliable set $\mathbf{X}_u$ learned by mutual information loss, the expected gradient estimation error on the whole dataset is:
$$
\begin{aligned}
& \mathbb{E} _{\mathbf{X}}[\mathbb{E}[\parallel \hat{\nabla} _{\boldsymbol{\theta}}{\check{\mathcal{L}} _i} - \nabla _{\boldsymbol{\theta}}\mathcal{L} _i \parallel _2]] \\\\ & = \mathbb{E} _{\mathbf{X} _r}[\mathbb{E}[\parallel \hat{\nabla} _{\boldsymbol{\theta}}{\check{\mathcal{L}} _i} - \nabla _{\boldsymbol{\theta}}\mathcal{L} _i \parallel_2]] + \mathbb{E} _{\mathbf{X} _u}[\mathbb{E}[\parallel \hat{\nabla} _{\boldsymbol{\theta}}{\check{\mathcal{L}} _i} - \nabla _{\boldsymbol{\theta}}\mathcal{L} _i \parallel _2]] \\\\ & = \mathbb{E} _{\mathbf{X} _r}[\mathbb{E}[\parallel \hat{\nabla} _{\boldsymbol{\theta}}\mathcal{L} _{\rm ce} - \nabla _{\boldsymbol{\theta}}\mathcal{L} _{\rm ce} + \hat{\nabla} _{\boldsymbol{\theta}}h - \nabla _{\boldsymbol{\theta}}h \parallel _2]] + \mathbb{E} _{\mathbf{X} _u}[\mathbb{E}[\parallel \hat{\nabla} _{\boldsymbol{\theta}}{\mathcal{L} _{\rm im}} - \nabla _{\boldsymbol{\theta}}\mathcal{L} _{\rm im} \parallel _2]] \\\\
& \leq \mathbb{E} _{\mathbf{X} _r}[\mathbb{E}[\parallel \hat{\nabla} _{\boldsymbol{\theta}}\mathcal{L} _{\rm ce} - \nabla _{\boldsymbol{\theta}}\mathcal{L} _{\rm ce} \parallel _2] + \mathbb{E}[\hat{\nabla} _{\boldsymbol{\theta}}h - \nabla _{\boldsymbol{\theta}}h \parallel _2]] + \mathbb{E} _{\mathbf{X} _u}[\mathbb{E}[\parallel \hat{\nabla} _{\boldsymbol{\theta}}{\mathcal{L} _{\rm ce}} - \nabla _{\boldsymbol{\theta}}\mathcal{L} _{\rm ce} \parallel _2]] \\\\
& \leq \mathbb{E} _{\mathbf{X}}[\mathbb{E}[\parallel \hat{\nabla} _{\boldsymbol{\theta}}\mathcal{L} _{\rm ce} - \nabla _{\boldsymbol{\theta}}\mathcal{L} _{\rm ce} \parallel _2] + \mathbb{E}[\hat{\nabla} _{\boldsymbol{\theta}}h - \nabla _{\boldsymbol{\theta}}h \parallel _2]].
\end{aligned}$$
>
> Thus, the upper bound of gradient estimation error is tightened after applying pseudo-label-robust data adaptation.

> Reference:
>
> [1] Boudiaf, Malik, et al. A unifying mutual information view of metric learning: cross-entropy vs. pairwise losses. ECCV, 2020.

---

### Decision · Program_Chairs · 2023-09-21

**Decision:**

Accept (poster)

**Comment:**

The proposed work on test-time data adapators for learning settings without access to model parameters is considered novel enough and initial concerns on theoretical and experimental justification on more realistic data have been sufficiently addressed in the discussion phase. Although the paper has no strong support, none of the reviewers is against acceptance. The AC has read the paper and after discussion with the SAC concludes that with the promised changes added to the camera-ready the work is acceptable. Adding more experimental results on non-synthetic datasets as commonly used in the test-time adaptation literature would strengthen the work even further.